# Training large-scale optoelectronic neural networks with dual-neuron optical-artificial learning

Xiaoyun Yuan [1,2,3], Yong Wang[1], Zhihao Xu[1,4], Tiankuang Zhou [1,2,3] & Lu Fang [1,2,3] ✉

Optoelectronic neural networks (ONN) are a promising avenue in AI computing due to their potential for parallelization, power efficiency, and speed. Diffractive neural networks, which process information by propagating encoded light through trained optical elements, have garnered interest. However, training large-scale diffractive networks faces challenges due to the computational and memory costs of optical diffraction modeling. Here, we present DANTE, a dual-neuron optical-artificial learning architecture. Optical neurons model the optical diffraction, while artificial neurons approximate the intensive optical-diffraction computations with lightweight functions. DANTE also improves convergence by employing iterative global artificial-learning steps and local optical-learning steps. In simulation experiments, DANTE successfully trains large-scale ONNs with 150 million neurons on ImageNet, previously unattainable, and accelerates training speeds significantly on the CIFAR-10 benchmark compared to single-neuron learning. In physical experiments, we develop a two-layer ONN system based on DANTE, which can effectively extract features to improve the classification of natural images.

The artificial neural network (ANN) is undoubtedly the most representative technology in the recent machine intelligence research field[1]. Over the past decade, with the growth of network scales[2], model parameters[3,4], and dataset sizes[5–7], ANNs have witnessed remarkable advancements in various fields, e.g., visual computing, natural language processing, robotics, etc. However, large-scale neural networks also placed tremendous pressure on existing electronic computing hardware. As the performance and energy efficiency of silicon-based computing devices are restricted by the plateauing of Moore's law[8], researchers started to turn their attention back to the optical/optoelectronic networks[9–12].

Optical and optoelectronic neural networks (ONN) have inherent high speed and high energy efficiency characteristics[13,14]. Among them, diffractive neural networks, which compute by just propagating encoded light through trained optical modulation elements, can naturally process optical images and realize the optical computing of various machine vision tasks[9,12,15–21]. In 2018, Lin et al. proposed the diffractive deep neural network (D²NN) for MNIST classification with 5 sequential 3D-printed masks and a Terahertz laser source. This idea is further extended for single-pixel imaging[22], optical linear transform[19,23], optical logic operation[24], phase imaging[25], saliency detection[16], etc. Liu et al. further propose a programmable D²NN based on a digital-coding metasurface array with adjustable network parameters[26]. Later, in order to realize stronger neural network processing capability, optoelectronic neural networks are proposed: the optical computing units are used to perform massively parallel linear operations, and the electronic computing units are used to multiplex the optical computing units and implement non-linear activation[9,27].

However, existing diffractive neural networks studies mainly focus on exploring novel optical computing hardware[23,26,28,29] or new

[1]Department of Electronic Engineering, Tsinghua University, Beijing, China. [2]Beijing National Research Center for Information Science and Technology (BNRist), Beijing, China. [3]Institute for Brain and Cognitive Science, Tsinghua University (THUIBCS), Beijing, China. [4]Tsinghua Shenzhen International Graduate School, Shenzhen, China. ✉e-mail: fanglu@tsinghua.edu.cn

network structures[16,18,20,30,31], but pay less attention to the modeling and optimization of the ONNs. Most of them accurately model the optical modulation elements as optical neurons with differentiable forward-pass functions. Then, they model the entire physical system to a large differentiable function by connecting all the optical neurons, and optimize all the parameters directly using backpropagation, called as single-neuron learning approach. However, the complex nature of optical diffraction modeling makes the differentiable functions of ONNs significantly more intricate and computationally demanding than those of ANNs. Consequently, this leads to a substantial optimization space and excessively long training times, posing significant challenges to the modeling and optimization of large-scale ONNs. Hence, most existing ONNs studies are still struggling with fundamental tasks and small datasets, e.g., MNIST and Fashion-MNIST classification.

In this article, we propose DANTE: dual-neuron optical-artificial learning for large-scale optoelectronic machine learning. In detail, the hardware network is modeled by both optical-neuron layers and artificial-neuron layers (Fig. 1). The optical-neuron layers accurately simulate the diffraction and modulation process of the optical field, and the artificial-neuron layers approximate the computationally heavy optical-diffraction modeling of the optical-neuron layers using lightweight functions. Unlike the single-neuron learning approach, DANTE decouples all the optical neurons by employing iterative global artificial-learning steps and local optical-learning steps. By introducing the artificial-neuron in the global artificial-learning step, the optimization space and computing memory requirement are significantly reduced, realizing faster and better convergence of the end-to-end network learning. While in the local optical-learning, the parameters in optical-neuron layers are learned independently and efficiently from the optimized artificial neurons, rather than from the massive datasets, which can further accelerate the network training. In simulation experiments, compared to the single-neuron learning approach, DANTE achieves a training acceleration of approximately 200 times and elevates accuracy by approximately 10% on the CIFAR-10 benchmark (Fig. 2). What is more, DANTE empowers the training of large-scale ONNs with 150-M neurons, achieving performance on par with the representative VGG network[4] on the modern ImageNet benchmark (Fig. 3). The

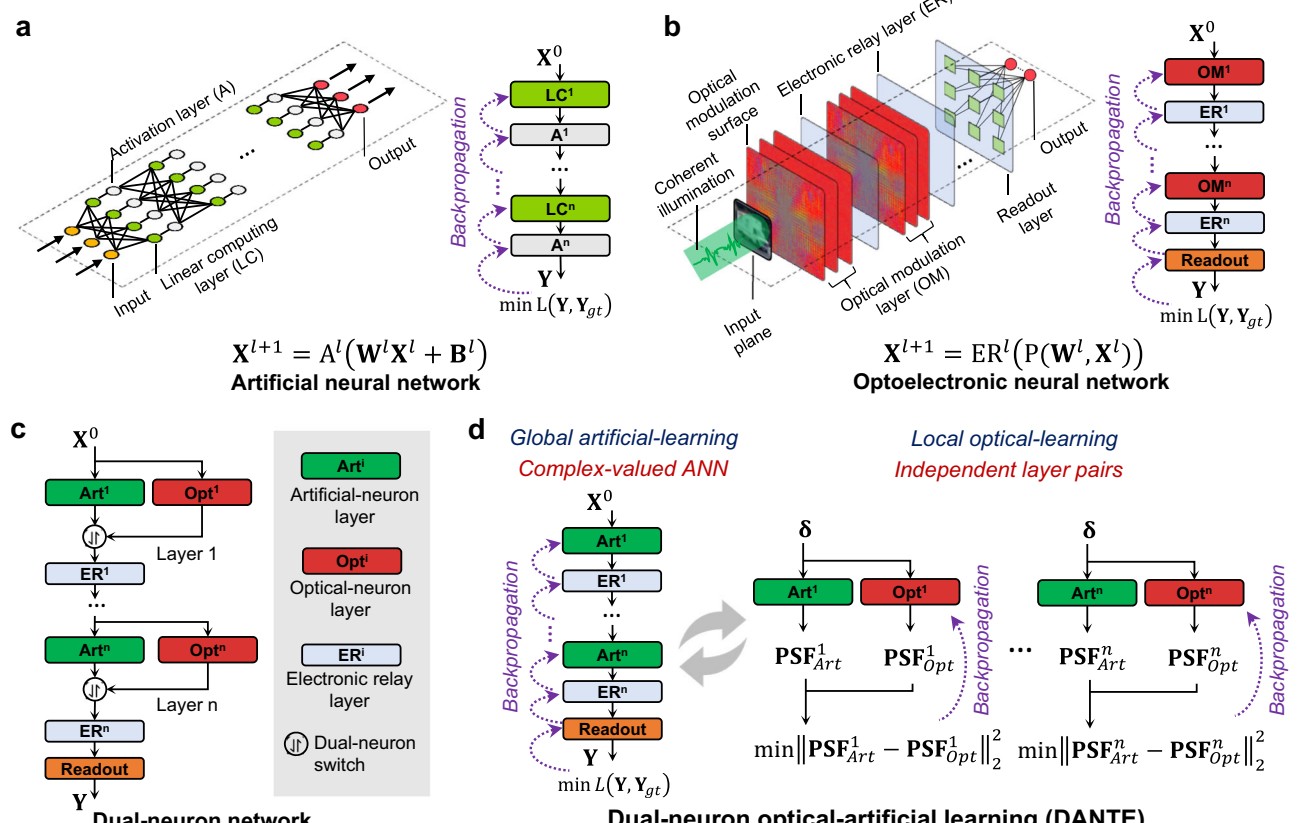

**Fig. 1 | Principle of dual-neuron optical-artificial learning (DANTE).**
**a**, **b** Schematic of an artificial neural network (ANN) and an optoelectronic neural network (ONN). These networks exhibit a similar architecture, comprising alternating layers of linear computing and non-linear activation. In the ONN, the optical modulation layer conducts linear transformation on the input, akin to the linear computing layer in the ANN, such as the convolutional layer. The electronic relay layer in the ONN corresponds to the activation layer in the ANN, introducing nonlinearity into the network. The ONN incorporates an additional readout layer to generate the final results. **c** our proposed dual-neuron network. Each optical modulation layer comprises parallel optical-neuron layer and artificial-neuron layer. The optical-neuron layer accurately models the optical diffraction based on Fourier optics. Meanwhile, the artificial-neuron layer aims to approximate the computationally intensive optical-neuron layer using easily-optimized computing operations. They are further connected via a dual-neuron switch, which dynamically adjusts the network structure during network training. **d** our proposed dual-neuron optical-artificial learning approach, composed of a global artificial-learning step and a local optical-learning step. In the global artificial-learning step, the dual-neuron switch connects all the artificial-neuron layers to form a complex-valued ANN, and backpropagation is employed to optimize the parameters of the connected artificial-neuron layers for fitting the training dataset. In the local optical-learning step, the dual-neuron switch decomposes the network into independent layer pairs, and the parameters in the optical-neuron layer is optimized by aligning its impulse response to closely match that of the corresponding artificial-neuron layer. $i$ denotes the index of the layer, $\delta$ is the Dirac delta function, **PSF**, point spread function, which represents the impulse responses.

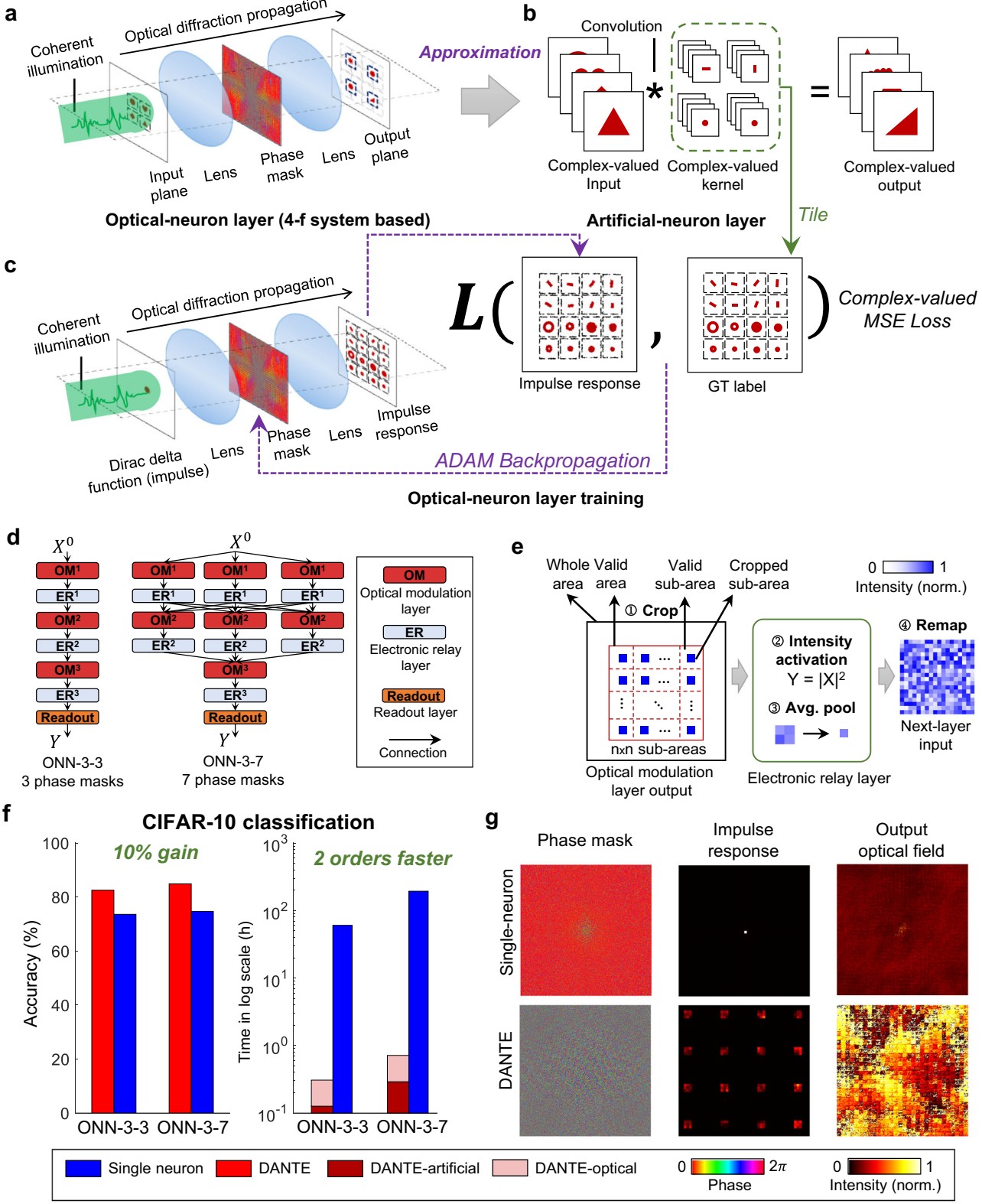

**Fig. 2 | Improving ONN training using DANTE. a** An optical-neuron layer, using a 4-f system as an example. **b** The artificial-neuron layer, which approximates the optical-neuron layer using a multi-channel complex-valued convolutional operation. **c** The training process of the optical-neuron layer. A 2D Dirac delta function (impulse) serves as the input, generating the 2D impulse response. The trained kernels in the artificial-neuron layer are then tiled in a plane, acting as the ground-truth label. ADAM-based backpropagation is employed to optimize the phase mask. **d** Two representative 3-layer ONNs for evaluating DANTE. **e** Detailed overview of operations conducted within the electronic relay layer (ER). **f** Classification accuracy and training time (log scale) comparison in simulation experiments. DANTE achieves over 100 times acceleration and a 10% accuracy improvement on CIFAR-10 dataset. **g** The trained phase masks, impulse response, and output optical field of the second optical modulation layer in ONN-3-3. The impulse response and output optical field are converted to intensity for visualization purposes. GT, ground truth.

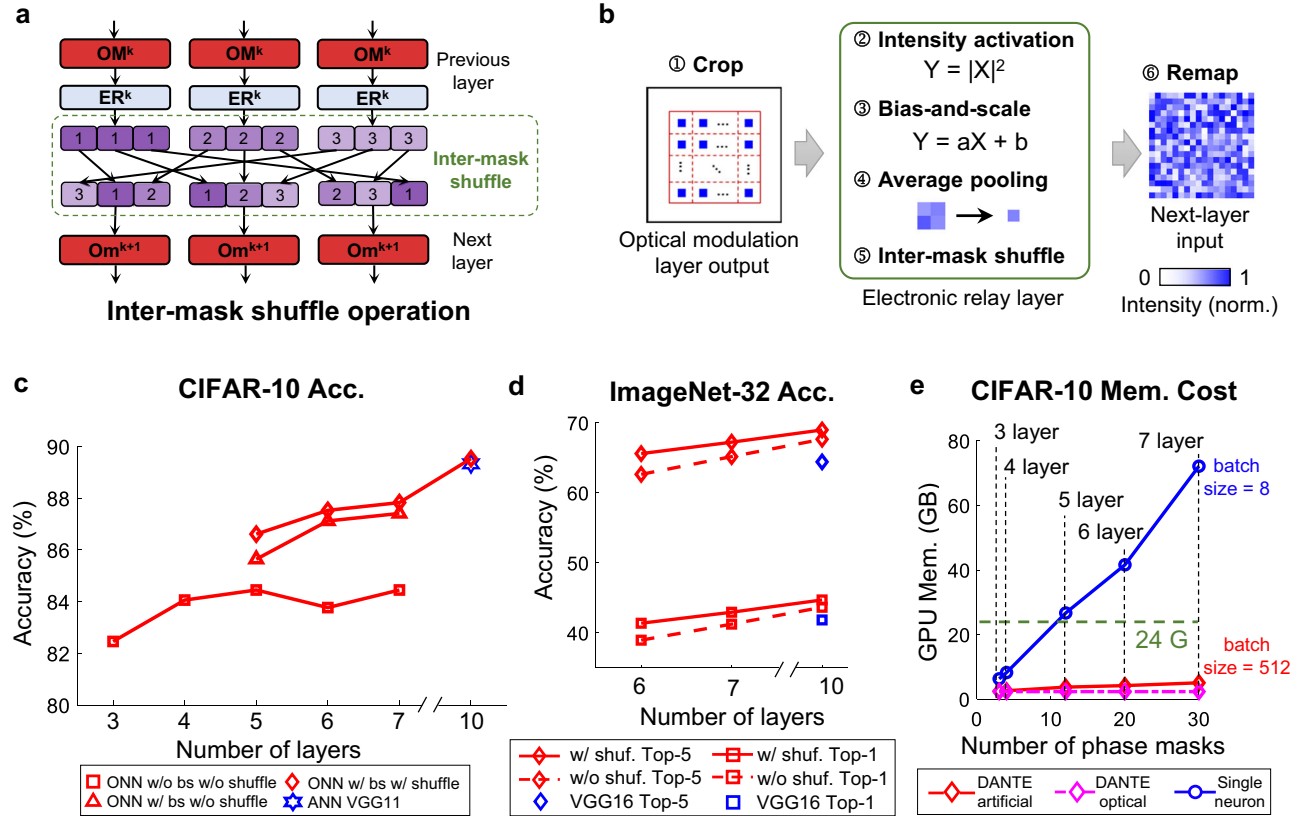

**Fig. 3 | Large-scale ONNs enabled by DANTE. a** The inter-mask shuffle operation. The outputs of previous layer are divided into multiple parts and shuffled to serve as the inputs for the next layer, similar to channel shuffle operation in ANN. **b** The electronic relay layer (ER) with two new operations: bias-and-scale and intermask shuffle, which are proposed to facilitate the training of larger and wider ONNs. **c** The classification accuracies of large-scale ONNs trained by DANTE on the CIFAR-10 dataset (in simulation). The VGG11 network is presented for comparison. **d** The Top-5 and Top-1 classification accuracies of large-scale ONNs trained by DANTE on the ImageNet-32 dataset (in simulation). The VGG16 network is presented for comparison. **e** GPU Memory cost for training large-scale ONNs on CIFAR-10 dataset. bs, bias and scale. Shuf., shuffle. Acc., accuracy. w/, with. w/o, without. Mem., memory.

network scale is approximately 10 times larger than existing large-scale ONNs[9,16,18,20,32]. In physical experiments, we develop a two-layer physical ONN system capable of effectively extracting features to enhance the classification of natural images (CIFAR-10 and ImageNet), serving as a validation of DANTE's physical feasibility (Fig. 4). In summary, DANTE shows the potential to advance the researches of optical computing from the early stage of prototype verification in MNIST-like benchmark into a new era of solving large-scale practical problems.

## Results

### Principle of dual-neuron optical-artificial learning (DANTE)

Figure 1a, b illustrates the schematics of an artificial neural network (ANN) and a diffractive optoelectronic neural network (ONN). The ANN is composed of alternating linear computing layers and non-linear activation layers. The forward pass of an ANN layer is usually formulated as follows:

$$\mathbf{X}^{l+1} = A^l\left(\mathbf{W}^l\mathbf{X}^l + \mathbf{B}^l\right) \qquad (1)$$

Where $\mathbf{X}^l$ denotes the input of the $l$-th layer, $\mathbf{W}^l$ and $\mathbf{B}^l$ represent the trainable weights and biases of the $l$-th linear computing layer, and $A^l(\cdot)$ is the activation function of the $l$-th nonlinear activation layer.

Similarly, the diffractive neural networks also consist of alternating linear optical modulation layers (OM layer) and non-linear electronic relay layers (ER layer), as demonstrated in Fig. 1b. The input information is first encoded to either the amplitude or phase of a

coherent optical field. The optical field then propagates through the layers until it reaches the final readout layer to generate the output. Mathematically, the forward pass of an ONN layer can be formulated as:

$$\mathbf{X}^{l+1} = ER^l\left(OM^l(\mathbf{X}^l)\right) = ER^l\left(P(\mathbf{W}^l, \mathbf{X}^l)\right) \qquad (2)$$

where $OM^l(\cdot)$ represent the $l$-th OM layer with trainable optical modulation parameters $\mathbf{W}^l$, and $P(\cdot)$ is the optical wave propagation function. $ER^l(\cdot)$ denotes the non-linear activation function of the ER layer.

More specifically, the OM layer is composed of multiple 2D optical modulation surfaces, and each pixel on the optical modulation surfaces functions as an optical neuron, enabling the manipulation of the optical field's amplitude[18] or phase[9,17]. In physical systems, the optical modulation surface can be realized using various approaches, including pre-fabricated devices like lenses or metasurfaces, as well as programmable devices like LCOS-Spatial light modulators (for phase and amplitude) and Digital micromirror devices (for amplitude). The primary purpose of the electronic relay layer is to introduce non-linearity. Among existing optoelectronic neural networks[9,18,27], the widely employed device for this layer is the photodiode array, which achieves nonlinearity by converting the complex optical field into a real-valued electrical voltage field. The voltage field can then be re-encoded back into a complex optical field, serving as the input of the subsequent layer.

In network learning, existing ONN approaches just follow the principles of ANNs, i.e., cascading all the layers to a large differentiable

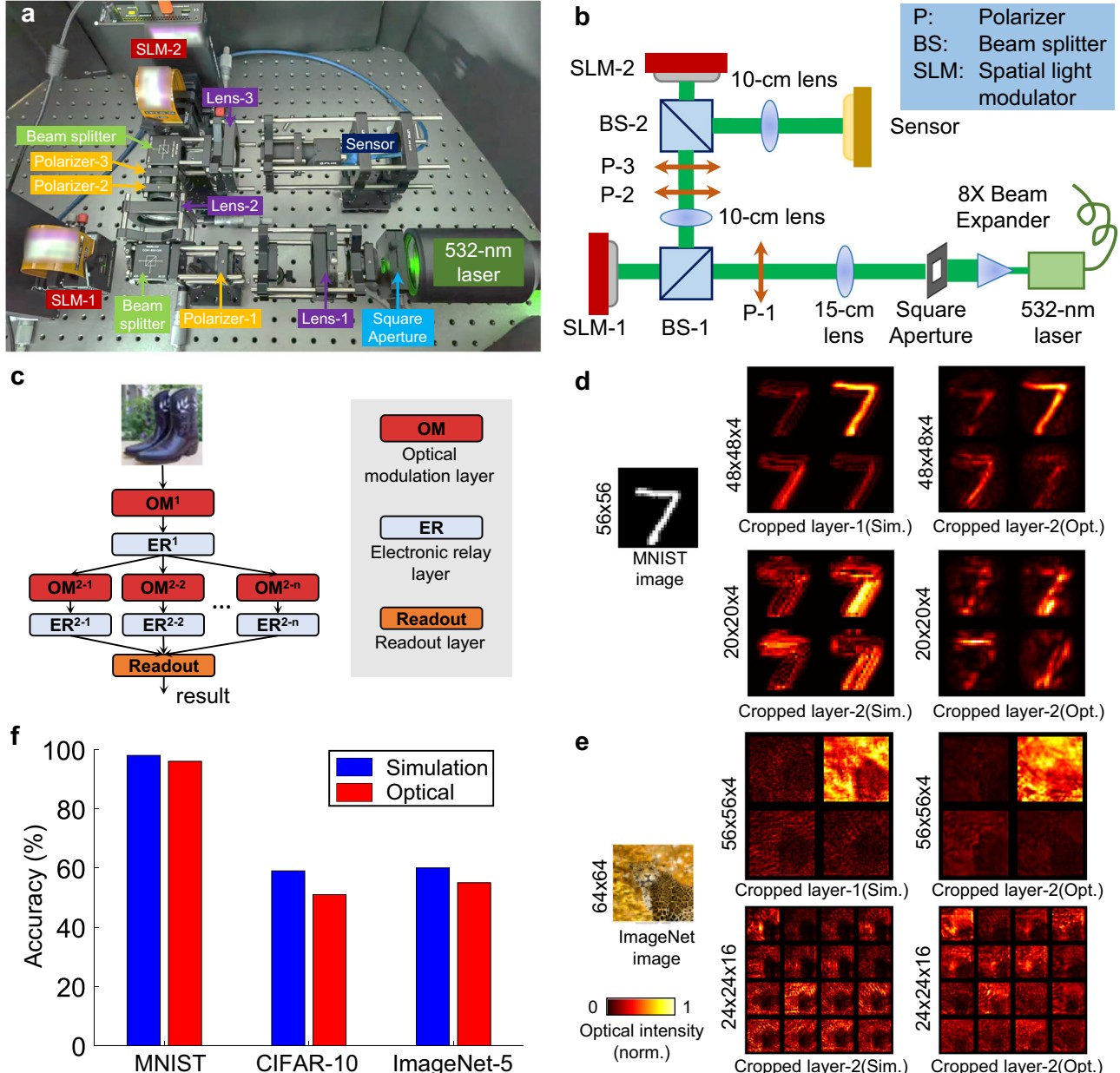

**Fig. 4 | DANTE on a physical ONN system. a, b** Prototype system and its optical diagram. A 532-nm laser and an 8X beam expander are used to generate flat coherent wavefront. A square-shape aperture is used to obtain a square-shape laser spot. A 15-cm lens (Lens-1) is used to shrink the square-shape laser spot, making it close to the size of the input image. SLM-1 is used to encode the input image, two polarizers (P-1 and P-2) are combined to set the SLM to amplitude modulation mode. Another polarizer (P-3) is used to make the polarization direction of the wavefront the same as the fast axis of the SLM-2. Two 10-cm lenses (Lens-2 and Lens-3) are used to form a 4-f system, and the SLM-2 is put at the Fourier plane of the system. A mono-color machine vision sensor is used to capture the output optical intensity. **c** The network structure implemented by the prototype system. **d** Cropped outputs of trained ONN for MNIST classification. Sim., simulation results. Opt., optical results generated by the physical experimental system. The number on the left denote the feature map size (pixel pitch: 8 μm). **e** Outputs of trained ONN for ImageNet classification. **f** Simulation and optical accuracy on MNIST, CIFAR-10, and ImageNet datasets. Sim., simulation. Opt., optical.

function ONN(·), and optimize all the trainable parameters using backpropagation:[9,12,16]

$$\min_{\mathbf{W}} \mathrm{Loss}\Big(\mathrm{ONN}\big(\mathbf{X}^0\big), \mathbf{Y}_{gt}\Big)$$

$$\mathrm{ONN}\big(\mathbf{X}^0\big) = \mathrm{Readout}\Big(\mathrm{ER}^n\Big(\mathrm{OM}^n\big(\ldots \mathrm{ER}^1\big(\mathrm{OM}^1\big(\mathbf{X}^0\big)\big)\big)\Big)\Big) \quad (3)$$

where n is the total number of layers, $\mathbf{Y}_{gt}$ is the ground-truth labels, Readout(·) the final readout layer. Unfortunately, this method faces two challenges in optimizing large-scale ONNs: convergence difficulty and slow optimization speed. For instance, the 3-layer ONN (D-NIN-1)

proposed by DPU[9] achieves comparable performance in MNIST classification compared to LeNet-5[33], but at a cost of 35 times more parameters (2.2 M vs. 62 K) and 700 times longer training time (8.4 h vs. 40 s). The challenges arise from the high computational cost associated with the modeling of optical diffraction propagation. Based on the angular spectrum (AS) method[34], the computation of wave diffraction between two surfaces requires two 2D Fast Fourier Transformations (FFTs) and one element-wise multiplication. Typically, an optical modulation layer may consist of multiple surfaces, necessitating multiple rounds of computations. In contrast, conventional ANN layers only require a single straightforward multiplication

operation. Consequently, the computational cost and convergence difficulty of an optical modulation layer is significantly higher than an ANN layer. For a more comprehensive analysis, please refer to the Supplementary Note S6 and Table S2.

Here, we innovate the dual-neuron optical-artificial learning (DANTE), which aims to address the computing challenge associated with the learning of large-scale ONNs and enable their applications to complex machine learning tasks. Figure 1c illustrates our dual-neuron ONN, where the optical modulation layer is modeled by a parallel dual-neuron structure, including an optical-neuron layer (represented as $Opt^i$), an artificial-neuron layer (represented as $Art^i$), and a dual-neuron switch. The optical-neuron layer accurately models the optical diffraction using the AS method. The artificial-neuron layer approximates the optical-neuron layer as an easily-optimized complex-valued linear computing function, aiming to reduce the computational cost. The optical-neuron layer and artificial-neuron layer are connected through the dual-neuron switch, responsible for changing the connectivity during the training process. In particular, Fig. 1d illustrates our dual-neuron optical-artificial learning approach, comprising two steps: global artificial learning and local optical learning. During the global artificial-learning step, the dual-neuron switch connects all the artificial-neuron layers to form a complex-valued ANN. Global backpropagation is then utilized to optimize the ANN to fit the training dataset. During the local optical-learning step, the dual-neuron switch divides the network into independent layer pairs, where each pair consists of an artificial-neuron layer and an optical-neuron layer. A loss function is applied to each pair to optimize the optical-neuron parameters using guidance from the trained artificial-neuron layer. Here, an impulse function acts as the input, where the output of the artificial-neuron layer serves as the training label. The objective is to optimize the impulse response of the optical-neuron layer to closely match that of the artificial-neuron layer. The objective function of global artificial learning is:

$$\min \mathrm{Loss}\left(\mathrm{ANN}\left(\mathbf{X}^0\right), \mathbf{Y}_{gt}\right)$$
$$\mathrm{ANN}\left(\mathbf{X}^0\right) = \mathrm{Readout}\left(\mathrm{ER}^n\left(\mathrm{Art}^n\left(\ldots \mathrm{ER}^1\left(\mathrm{Art}^1\left(\mathbf{X}^0\right)\right)\right)\right)\right) \quad (4)$$

The objective function of local optical learning:

$$\min ||\mathrm{Art}^i(\boldsymbol{\delta}) - \mathrm{Opt}^i(\boldsymbol{\delta})||_2^2 \quad (5)$$

where $\mathrm{ANN}(\cdot)$ is the forward function of the complex ANN, $\boldsymbol{\delta}$ denotes the Dirac delta function (impulse), $\mathrm{Art}^i(\cdot)$ and $\mathrm{Opt}^i(\cdot)$ denote the forward functions of the i-th artificial-neuron layer and optical-neuron layer, respectively. In contrast to existing approaches that directly optimize the optical-neuron layer to fit the training dataset, our approach simplifies the process by fitting the clean Dirac delta functions. This optimization strategy leads to a remarkable reduction in the computational complexity during network training. For a comprehensive understanding, please refer to Supplementary Note S2 and Fig. S2.

### Improving ONN training using DANTE

Figure 2a–c demonstrate the training process of the optical-neuron layer, using a 4-f system as an example. Initially, the input is encoded into a coherent optical field, which propagates through two lenses and a trainable phase mask, producing outputs on the output plane. Based on Fourier optics, the 4-f based optical-neuron layer can be approximated as a complex-valued convolution operation with a large-size input plane and kernel. To further reduce the computational cost, it is decomposed into a multi-channel complex-valued convolutional operation with small-size inputs and kernels, serving as the forward function for the corresponding artificial-neuron layer. As the convolution operation is linear shift-invariant (LSI), the optical-neuron layer can be fully characterized by its 2D impulse response. The

optimization process is demonstrated in Fig. 2c. A 2D Dirac delta function (impulse) is used as the input, and the resulting output is compared to the ground-truth label, which is obtained by arranging the multi-channel complex-valued kernels into a tiled large-size single-channel complex-valued plane. ADAM-based backpropagation is employed to learn the optical modulation parameters. For more details on how the kernels and inputs are tiled, as well as the specifics of optimizing the optical modulation parameters, please refer to Supplementary Note 1 and Fig. S1.

We conducted DANTE analysis on two representative ONN structures, ONN-3-3 and ONN-3-7, as depicted in Fig. 2d. ONN-3-3 is a 3-layer network comprising three optical modulation layers, three electronic relay layers, and one readout layer. ONN-3-7 draws inspiration from DPU[9], which is also a 3-layer network but with the first two layers having three parallel optical modulation layers. The 3 optical modulation layers in the second layer are multiplexed for the 3 outputs of the first layer. Figure 2e demonstrates the details of an electronic relay layer, encompassing sequential operations of cropping, intensity activation, average pooling, and remapping (Fig. 2d). The readout layer consists of a small fully connected (FC) layer for inferencing the final output. Please refer to Supplementary Fig. S1 for the detailed network structures of ONN-3-3 and ONN-3-7. CIFAR-10[35] dataset is utilized for evaluation. Existing implementations[17,18] have not achieved acceptable performances on CIFAR-10 yet either in physical or in simulation. In addition to the accuracy, the extremely long training time is another non-negligible issue. Figure 2f demonstrate that DANTE makes dramatic advances in accuracy (in simulation) and training speed. Compared to the existing approach single-neuron learning approach, both the accuracies of ONN-3-3 and ONN-3-7 are improved by approximately 10% (ONN-3-3: 82.53% vs. 73.61, ONN-3-7: 84.91% vs. 74.67%), and the training speed is accelerated by more than 100 times (ONN-3-3: 0.3 h vs. 60 h, ONN-3-7: 0.7 h vs. 194 h). Compared to the existing ONN implementation[18], DANTE achieves ~20% improvement in simulation experiments (84.86% vs. 63%). Figure 2g illustrates the trained phase masks, impulse response, and output optical field of the second optical modulation layer in ONN-3-3, providing an explanation for the performance enhancement achieved by DANTE. As the phase masks are randomly initialized before training, the initial impulse response would be a bright point positioned at the center of the plane. In the single-neuron approach, optimization is primarily concentrated in the central-region pixels of the phase mask, leaving the remaining regions close to noise. Consequently, the impulse response remains similar to its initial value (a bright point). In contrast, DANTE can effectively optimize the whole phase mask, leading to the emergence of clear kernel patterns in its impulse response. The right column of Fig. 2g displays the output optical field when a CIFAR-10 image is used as the input. In the single-neuron approach, only the central region contains meaningful information, while the rest of the field approaches zero intensity. Conversely, in the results obtained by DANTE, meaningful image patterns fill the whole output plane, indicating a comprehensive and robust optimization process.

Collectively, compared to existing training approaches, ONNs trained by DANTE generates more effective information on the output plane, leading to a remarkable accuracy gain of approximately 10% on the CIFAR-10 classification task. Moreover, DANTE significantly accelerates the training speed of ONNs, outperforming existing methods by more than 100 times.

### Large-scale ONNs enabled by DANTE

In this section, we demonstrate that DANTE has the potential to facilitate the training of large-scale ONNs that were previously impossible to train. Research in the field of ANN has established the significance of network width and depth for improving performance[36,37]. However, diffractive neural networks face difficulties in expanding network width due to the physical size constraints of phase masks. In order to

solve this, we first employ multiple parallel optical modulation layers (OM) within the same layer to increase the number of trainable parameters. Then, we add the intermask shuffle operation in the subsequent electronic relay layer (ER) to establish connections among these parallel OM layers (Fig. 3a). Drawing inspiration from the batch normalization technique in ANN, we incorporate the Bias and scale operation in ER to facilitate the training of much deeper ONNs (Fig. 3b).

To evaluate the performance of DANTE, we utilize the CIFAR-10 dataset and train ONNs with increased depth and width. We train six ONNs with 3, 4, 5, 6, 7, and 10 optical modulation layers, corresponding to 3, 4, 12, 20, 30, and 63 trainable phase masks, respectively (Supplementary Fig. S4). The accuracies obtained through Fourier optics-based simulation are presented in Fig. 3c. The red curve with square markers represents ONNs without bias-and-scale and shuffle, which exhibit a plateau in accuracy after 5 layers. Conversely, the red curve with triangle markers represents ONNs with bias-and-scale but without shuffle, showcasing continued accuracy improvement beyond 5 layers. The 7-layer ONN experiences an approximately 4% increase in accuracy. Furthermore, the introduction of the shuffle operation further enhances accuracy (red curve with diamond markers). We also construct a larger ONN with 10 optical modulation layers and 1 readout layer in simulation experiments, which is comparable to the representative VGG11 network (89.53% vs. 89.33%).

To further push the boundaries of ONN performance, we conducted experiments on the more challenging ImageNet-32 dataset (Fig. 3d). Similarly, we train 3 ONNs with 6, 7, and 10 optical modulation layers, corresponding to 48, 64 and 104 trainable phase masks, respectively (Supplementary Fig. S5). For comparison, VGG16[4] network with 13 convolutional layers and 3 fully-connected layers is selected. The impact of shuffle operation on the CIFAR-10 dataset is relatively minor (around 0.6%), since the ONNs used for the CIFAR-10 dataset are not exceptionally wide. While in the ImageNet-32 classification, the shuffle operation led to a notable 2% improvement in accuracy. Remarkably, our 10-layer ONN in simulation achieves a top-1 accuracy of 44.26% and a top-5 accuracy of 68.61%, comparable to the VGG16 network (41.85% top-1 accuracy, 64.42% top-5 accuracy).

Figure 3e provides a clear illustration of the spatial complexity involved in training these ONNs on the CIFAR-10 dataset. The single-neuron approach models the entire network within the memory and optimizes all the phase masks simultaneously. As depicted by the blue curve, the memory cost rises in proportion to the number of trainable phase masks. As a result, training a 5-layer ONN with 12 phase masks at a batch size of 8 would exceed the memory capacity (24 GB) of a single GPU. In contrast, the artificial-neuron layer dramatically reduces spatial complexity, allowing the network to be trained with little memory usage (<6GB) even at a large batch size of 512. In the case of local optical learning, where only one phase mask needs to be trained at a time, the memory cost keeps constant at approximately 2.5 GB. Furthermore, the significant spatial complexity also implies a substantial temporal complexity. For instance, on the CIFAR dataset, training a phase mask for one epoch takes approximately 900 seconds for the single-neuron approach. Consequently, training the 7-layer ONN with 63 phase masks would require an average of 16 hours per epoch, and completing the entire network training would require several weeks. However, with DANTE, we can remarkably shorten the training process to a mere several hours. For more details, please refer to Supplementary Note S4 and Supplementary Table S1.

To summarize, DANTE significantly reduces the spatial and temporal computational complexity associated with training ONNs. This breakthrough allows for the training of deeper and wider ONNs using existing GPUs within a feasible amount of time. Through our demonstrations, we demonstrate that, with proper optimization, the large-scale ONNs have the potential to achieve comparable performance to existing ANNs on modern large-scale datasets.

## DANTE on a physical ONN system

We develop a custom ONN system using off-the-shelf optical modulation devices to verify the physical feasibility of DANTE (Fig. 4a, b). The system implements the optical computing function of an optical modulation layer. The input is modulated to the SLM-1, the network parameters are modulated to the SLM-2, and the CMOS sensor is used to receive the computing results. See Methods for the detailed specifications of the ONN system. Similarly, the MNIST, CIFAR-10, and ImageNet benchmarks are used to evaluate the ONNs' performance. Please refer to the Methods and Supplementary Note S5 for the preparation of the datasets and the detailed experimental steps. The implemented two-layer ONN structure is presented in Fig. 4c. The first layer only has one optical modulation layer, and the second layer has several parallel optical modulation layers. The outputs of the second layer are input to the readout layer to predict the final results. See Supplementary Fig. S10 for detailed network structures, and refer to the Methods section for the more details of the physical system, the implementation of data preprocessing and the baseline approach.

Figure 4d demonstrates the outputs of an MNIST sample 7. The optical intensity maps captured by the sensor (opt.) show a similar distribution to the simulation results (sim.). The center zero-order diffraction can be removed using an intensity correction mask (Supplementary Fig. S9). The difference between the simulation results and optical results (output of the ONN system) mainly comes from the imperfect coherent wavefront and the millimeter-level assembly error of the optical modulation devices. Hence, we re-tune the FC layer in the readout layer to compensate the errors. Figure 4e presents the outputs of the ImageNet-32 dataset (a leopard image as an example). Compared to the MNIST results, the differences between the simulation and optical results become larger due to the more complex image contents, but we can still see similar optical intensity distributions. The optical results are blurrier than the simulation results, which is also caused by assembly errors and system noise. Please refer to Supplementary Movie 1 and Supplementary Movie 2 for more results generated by our physical ONN system.

Figure 4f shows the quantitative analysis results. For simple binary-like MNIST dataset with a clean background, DANTE achieves ~96% accuracy, 2% below the full simulation results. In terms of training time, the global artificial-learning step achieves convergence in 60 epochs, requiring approximately 135 seconds, while the local optical-learning step demands approximately 280 seconds for optimizing two-phase masks. The re-tuning of the FC layer costs about 30 seconds. Taken together, the entire training process is completed within approximately 445 seconds. Notably, when comparing to existing single-neuron learning approaches like DPU[9], which takes over 5 hours to train on the MNIST benchmark, our method significantly accelerates the training process. For natural images with a complex background, the gap between the simulation and optical results become larger, about 8% for the CIFAR-10 dataset (59% vs. 51%) and 5% for the 5-class ImageNet dataset (60% vs. 55%). Although there is still a gap between the simulated ONNs and the physical ONNs due to the intrinsic error and noise in the physical system, the optical results still significantly outperform the baseline method (linear classifier), which proves that our physical ONNs can effectively extract features from the input images. The experimental results on the ONN system validate that the physical feasibility of DANTE. Looking ahead, there exists the exciting potential to integrate the physical ONN system with high-precision nanofabrication techniques, which could significantly elevate its computational capabilities.

## Discussion

This study introduces DANTE, a dual-neuron optical-artificial learning approach, and showcases its remarkable advancements in training large-scale ONNs. Importantly, we provide evidence for the first time

that wave-based ONNs can achieve comparable performance to modern ANNs in simulation, such as VGG, on modern large-scale datasets like ImageNet. We also provide evidence that DANTE can be successfully implemented on physical ONNs, as depicted in Fig. 4. The optical modulation layers can be constructed using readily available off-the-shelf optical modulation devices or customized devices developed using lithography technology[17]. Currently, the computations of the electronic relay layer (ER) are performed on a PC. In the future, these computations can be efficiently implemented by customizing photodetector arrays. Specifically, the intensity activation operation (complex to real conversion) can be automatically completed by the photodiode, while the crop and average pooling operations can be achieved by designing the photodiode pixels with appropriate size and position, and the bias-and-scale operation can be accomplished by configuring specific readout circuits.

The issue of optical-artificial fitting error is also worth discussing. In our setup based on 4-f system, the phase mask plane corresponds to the frequency spectrum, where a larger trainable phase mask size indicates a higher modulation frequency and finer impulse response. Through careful design of the size and number of impulse responses in the artificial-neuron layer, the fitting error would be within acceptable limits. Please refer to Supplementary Fig. S7 for more details.

Evaluating the computational performance, energy efficiency, and inference time of ONNs is a common practice in the literature. Existing studies have also discussed these aspects[9,18]. Trillions of Operations Per Second (TOPS) is a simplifying metric for measuring the computing performance of AI accelerators, which is also widely used for ONN studies[9,38]. Based on Fourier optics, the 4-f system is equivalent to two Fast Fourier Transformation (FFT) operations and one element-wise multiplication operation. For an $N \times N$ optical field, one FFT requires $5N^2\log_2 N^2$ real-valued operations (OPs), and the complex-valued element-wise multiplication takes $6N^2$ real-valued OPs. Hence, the optical modulation layer involves $10N^2\log_2 N^2 + 6N^2 = 9 \times 10^8$ real-valued operations ($N = 2000$). The inference time of our system is primarily limited by the speed of the SLM and the sensor, as the light propagation time is negligible. Currently, the SLM can achieve a framerate of 1.4 KHz[39] and the super high-speed camera can capture images at approximately 2000 FPS with 1 K resolution[40], resulting in an inference time of ~0.7 ms for one optical modulation (OM) layer. In our prototype system, the laser, SLMs, sensor and control computer require about 65 Watts in total. Therefore, the computing efficiency of our prototype ONN system is $9 \times 10^8\ OPs \times 1400\ \text{FPS}/65\ \text{W} = 0.02$ TOPS/W (analog real-valued operation). Note that there remains substantial room for improvement by implementing ONNs on photonic chips. For comparison, the computing performance of NVIDIA RTX 3090 GPU is 35.58 TOPS (32-bit float-point operation) with a thermal design power (TDP) of 350 Watt, resulting in a computing efficiency of 0.1 TOPS/W (32-bit float operation). Interestingly, the commonly used TOPS metric may not always accurately reflect the actual performance. For example, completing two FFTs and one element-wise multiplication operations on an RTX 3090 GPU costs approximately 0.9 ms, slower than our ONN prototype system (0.7 ms). Moreover, different computing techniques may employ different methods for counting the number of operations (OPs). This variation can also significantly impact performance evaluations. For more detailed discussions, please refer to Supplementary Note S6.

While we currently focus on demonstrating the performance of DANTE on 4-f system-based diffractive neural networks, the underlying concept can be extended to other types of diffractive neural networks[12] as well. The implementation details and results regarding this extension are presented in Supplementary Note S3 and Fig. S3. In the future, we can further broaden the applicability of our method to encompass other types of ONN architectures, such as on-chip diffraction-based ONNs[41] and integrated chip diffractive neural network[42]. Through the utilization of optical-artificial dual neurons in modeling

ONN chips, we have the potential to expedite network training and increase the size of trainable networks.

In conclusion, our dual-neuron optical-artificial learning (DANTE) framework effectively tackles the learning challenges faced by diffractive optoelectronic neural networks (ONNs), which arise from the intricate spatial and temporal complexities involved in optical diffraction modeling. Consequently, we have achieved remarkable success in training large-scale ONNs that were previously considered impossible-to-train using existing approaches. The experimental results demonstrate the enormous potential of ONNs in advanced machine vision tasks. We firmly believe that our research will establish a solid theoretical foundation for the training and deployment of large-scale ONNs, paving the way for a new era in which ONNs can solve large-scale practical problems.

## Methods

### Optoelectronic neural networks (ONN) settings and implementation

The spatial simulation size of the optical modulation layer is set to $16 \times 16$ mm with a resolution of $2000 \times 2000$ pixels (8-μm pixel pitch) for the input plane, lens plane, phase mask plane, and the output plane. In the phase mask plane, only the center $1200 \times 1200$ pixels are trainable (trainable phase mask size), while the light outside this region will be blocked. The wave propagation model is derived based on the angular spectrum method[34] (AS). The large spatial size ($2000 \times 2000$) ensures the boundary conditions of the simulation. The pixel pitch used in the simulation matches that of the commercial Liquid-crystal-on-silicon (LCOS) SLMs. Both the simulation and physical experiments employ a 532-nm laser. For simulation ONN (Figs. 2, 3), the focal length of the lenses is set to 14.5 mm to strike a balance between the frequency modulation resolution and range. In the physical ONN system, two 10-cm lenses are used to form a 4-f system.

In global artificial-learning step, an ADAM optimizer is used for MNIST and CIFAR-10 dataset, and an SGD optimizer is used for ImageNet32 dataset. Cross entropy loss is used to train all the connected artificial-neuron layers. In local optical-learning step, an ADAM optimizer is used for gradient calculation, and a MSE loss is applied on the valid region of the output complex impulse response map to train the phase modulation parameters. The network training is implemented using the PyTorch framework version 1.11.0 (Meta AI) running on a Linux server (Nvidia RTX 3090 GPU, Intel Xeon Gold 6248 R CPU with 96 cores, 256 GB of RAM, and the Ubuntu 18.04.6 LTS operating system).

### Physical ONN system

A continuous 532-nm laser (MGL-FN-532, Changchun New Industries Optoelectronics Technology Co., Ltd) and an 8X beam expander (BEF08-A, 8X, Shenzhen LUBANG Technology Co., Ltd.) are used to generate flat coherent wavefronts. Two Liquid-crystal-on-silicon SLMs (E-Series 1920 × 1200, Meadowlark Optics Inc., USA) are used to modulate the phase of the wavefronts. The SLMs are calibrated at 532 nm, with an 8-μm pixel pitch size and 1920 × 1200 resolution, and controlled by the HDMI port. A grayscale CMOS camera (Blackfly S BFS-U3-51S5M-C, FLIR LLC, USA) is used to capture the output optical intensity field. The camera resolution is 2448 × 2048 with a pixel pitch size of 3.45 μm. Two 10-cm lenses are used to form a 4-f system. The first SLM and the sensor are put at the input and output plane of the 4-f system. The second SLM is put at the Fourier plane to modulate the phase of the input optical wavefront. The SLMs are controlled using the Meadowlark MATLAB SDK, and the camera is controlled using python scripts.

### Dual-neuron layer modeling and optimization

Based on Fourier optics, a 2-f system consisting of two propagations stages and one lens can be effectively modeled as a Fourier transform

operation, and the 4-f system can be equivalently represented as a complex-valued convolution operation. The input images and convolution kernels for this complex-valued convolution operation only have a single channel, but their spatial dimensions are large, the same as the spatial simulation size. $31 \times 31$ is considered a large-size convolutional kernel, while the complex-valued kernel here is tens of times larger in size[43]. However, based on existing convolutional neural network (CNN) studies, large-size kernel is difficult and computational expensive to train, and its improvement is relatively small compared to the costs. Representative CNNs prefer small-size kernels like $3 \times 3$ and $5 \times 5$. Therefore, we decompose the single-channel large-kernel complex-valued convolutional operation into a multi-channel complex-valued convolutional operation with small-size inputs and kernels, which can further reduce the computational cost and memory requirement. Specifically, the multi-channel input is tiled into a single-channel complex-valued matrix, which serves as the input of the optical-neuron layer. The multi-channel complex-valued kernel is first zero-padded to the same size as the multi-channel input, and then tiled into a single-channel large kernel to serve as the ground-truth label for optimizing the phase mask. When convolution is done between the tiled input and tiled kernel, the regions containing the output information will be cropped from the output plane and used by the following electronic relay (ER) layer. Please refer to Supplementary Note S1 and Fig. S1 for more detailed discussion.

### Dataset preprocessing and baseline approach of physical ONN system

The CIFAR-10, ImageNet-32 datasets are used for performance evaluation in simulation experiments. The MNIST, CIFAR-10, and ImageNet-32 are used for evaluating the physical ONN system.

1. In all the simulation experiments, we use the original training and testing samples of these datasets without modification. Data augmentation including Normalize, RandomHorizontalFlip, RandomErasing and RandomCrop is used to reduce the overfitting.

2. In physical experiment of MNIST dataset, we use the first 10000 images in the training set for training, and the whole test dataset (10000 images) for testing. For better image quality, we resize the original $28 \times 28$ image to $56 \times 56$ in experiments. As the MNIST images are relatively simple, using only a part of the training set is enough to achieve a very high accuracy.

3. In physical experiment of CIFAR-10 dataset, we use the whole training dataset (50,000 images) for training, and the whole test dataset (10,000 images) for testing. Compared with MNIST, CIFAR-10 images are more diverse, so all the images of the training set are used.

4. In the physical experiment of the large-scale ImageNet-32 dataset (over 1 M images), we choose 5 classes from the original 1000 classes to form a small subset to evaluate our system. Each class has 1300 images and we use the first 1000 images for training and the last 300 images for testing. In total, we have 5000 training sample and 1500 testing samples. As the number of training images for each class is small, we use data augmentation to reduce the overfitting.

In the physical experiments, we construct the baseline accuracy using a linear classifier and cross-entropy loss: the original 2d images are flattened into vectors and input to a fully connected (FC) layer to output the classification results. The baseline accuracy of the MNIST, CIFAR-10, and ImageNet-32 datasets is 91%, 37%, and 45%. While the accuracy of our physical ONN system is 96%, 51%, and 55%, which is much higher than the baseline method.

### Data availability

The datasets used in this study are publicly available and can be accessed through the following sources. The MNIST dataset used in this study can be obtained at https://www.kaggle.com/datasets/hojjatk/mnist-dataset. The CIFAR-10 dataset used in this study is available from the Canadian Institute for Advanced Research (CIFAR) and can be accessed at https://www.cs.toronto.edu/~kriz/cifar.html. The ImageNet dataset is available through the ImageNet project at https://image-net.org/download-images.

### Code availability

Code and instructions to reproduce the results are available at the GitHub repository https://github.com/yuanxy92/DANTE.

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

## Acknowledgements

L.F. acknowledges support from Ministry of Science and Technology of China under contract No. 2021ZD0109901, Natural Science Foundation of China (NSFC) under contract No. 62125106, 61860206003, 62088102, and Tsinghua-Zhejiang joint research center. X.Y. acknowledges support from Natural Science Foundation of China (NSFC) under contract No. 62271283 and Young Elite Scientists Sponsorship Program by CAST No. 2021QNRC001. T.Z. acknowledges support from Shuimu Tsinghua scholar program.

## Author contributions

L.F. initiated and supervised this study. X.Y. and Y.W. conceived the research and method. X.Y. and Z.X. build the ONN system and perform the experiments. X.Y., Z.X., and T.Z. wrote the manuscript. All authors discussed the research.

## Competing interests

The authors declare no competing interests.
