## [Peer Review File · Nature Communications]

REVIEWER COMMENTS

Reviewer #1 (Remarks to the Author):

I find the objective of the manuscript very interesting –if I have understand correctly also the technique employed by the authors. I have reviewed the manuscript at an earlier stage, and I must say that the authors have done some effort to improve the readability. But it is still very cryptic. It is laden with a big amount of results, different systems, numerous repetitions of the concept on a superficial level and its advantages. I think I have grasped the method, but the authors still use convoluted concepts like switches etc. All of this is done on 2 pages, constantly referencing why the here presented approach is better, which is already stated in the introduction, the same section and through the rest of the manuscript at various times. Why don't you just explain concise what you do on half a page, then explain why its better and than focus on specific results. As it is I would suggest the authors rewrite the manuscript and focus on the physics and interesting idea.

Below please find some additional comments, but currently there are too many things at odds with the manuscript for a detailed discussion of the text and results.

Physically relevant details are almost entirely left out or mentioned very late, and instead the very most basic part of the principle idea is repeated again and again. There are now 2 equations describing the concepts, but the conceptual part is still very weak in its description. You: optimize a fully abstract and virtual ANN to perform a task in the first stage, then you optimize an optical system to mirror that abstract ANN in a second stage, and these stages are carried out for each learning epoch. Correct or not?

Reviewer #2 (Remarks to the Author):

In this manuscript, the authors described a training framework for optical neural networks (ONNs). Instead of training the ONN using a digital model that simulates the physical process of diffraction, they introduced an artificial neural network (ANN) to model exactly the same forward computation to assist with training. They first trained the ANN, since it involves fewer parameters, and then used the trained ANN as the reference to training the physical model of ONN in a layer-by-layer fashion, essentially by matching the impulse response of each layer of ONN and ANN. Compared to direct training on the physical diffraction model, this method probably improves training by reducing the

computation of each layer to alleviate the vanishing gradient problem, as the authors show in Fig. 2e and the associated discussion.

The manuscript tries to address an interesting and important problem, which is to find a more efficient and reliable way to train the hardware of diffractive neural networks, which at this stage still cannot be fully faithfully simulated on a computer to account for intrinsic noise and error in an actual physical setup. Despite the proposed idea being an interesting one, the work does not present enough evidence to support for the claim.

My primary concerns are as follows:

1. It seems to me that the proposed training method would work, and it is an improvement from the direct simulation of the physical process of diffraction in terms of computation time and training effectiveness. However, most ONNs, including the 4f convolution setup discussed extensively in this work (e.g., Fig. 1a), do not require simulation of optical diffraction to start with, and this is only a specific technical challenge with diffractive neural networks. The authors argued in a footnote that “the frequential resolution of the FFT result may be different from the pixel pitch of the physical devices”, and therefore the 4f setup cannot be efficiently modeled by FFT, but has to be modeled by diffraction. I am not fully convinced by the statement, since numerous prior works are training 4f convolutional neural networks without directly simulating optical diffraction, for instance, Ref. 19, Ref. 18 (which is the same as Ref. 36 by mistake). To really make this point, the authors should compare their method to other methods once used to train 4f systems. To be honest, in machine learning, there is no absolute criteria on which training method is better or worse, and a lot of times, it depends on the training hyperparameters and suitability to the application. Therefore, I suggest authors focus on explaining how the training schemes solve a previously impossible-to-solve training problem or how it might be helpful for other applications.

2. There is a lack of explanation on a very important step of the proposed training which is to compare the impulse response of an ANN layer to a delta function with that of an ONN layer. What training algorithm was used to optimize the parameters of the ONN layer? How many different delta functions are actually needed (despite the footnote saying “multiple”) Again, I strongly recommend the authors share their code online so that such details can be assessed from the reviewers’ point of view.

3. The manuscript claims “superior” performance on ImageNet and CIFAR-10 over digital models such as VGG. I support demonstrating ONNs on complex tasks performed by state-of-the-art digital models. However, the claim is based solely on simulation, and as far as I can see from the information provided, the simulation does not consider noise or other intrinsic errors. The demonstration of an experimental setup was only mentioned on line 271 and line 272. Most recent digital NN models can achieve well above 90% accuracy on CIFAR-10 (>99% is not rare either), while

the study achieves 51% test accuracy. Therefore, I think it is unfair to claim ONNs achieve better performance than digital NNs on hard tasks. For all similar claims in the paper, I would like to see compelling evidence where an experimental demonstration achieves higher test accuracy in CIFAR-10 or ImageNet compared to state-of-the-art digital models, and some explanation on why before I can accept such “superiority” claims.

4. The manuscript extensively discusses the number of parameters and the number of operations of the models. However, it does not mention the equivalent bit precision achieved by optical computing units. It is only fair to compare the number of parameters and operations when the bit depth (or precision in the case of analog computing) of the parameter and operation is known. Therefore, I do not find these comparisons quite insightful or contributing to the study.

Once more, I think the key message this manuscript tries to convey is a more robust and efficient training method for diffractive neural networks. I think the method proposed by the authors worked well in several aspects compared to the direct simulation of the physical process of diffraction; however, instead of focusing on explaining how this method might be useful for, the authors misplaced emphasis on several irrelevant claims that are not as solid if you think carefully about them. Nevertheless, this is still a good piece of work with some potential, and I would like to hear the response from the authors.

Reviewer #3 (Remarks to the Author):

This paper presents a number of exciting results in a multi-faceted field using an innovative training technique. Training optoelectronic neural networks (ONNs) is a challenging task that demands careful solutions. While employing more conventional artificial neural network (ANN) neurons in tandem with optical neurons is good idea, it may be appropriate to more directly state that this is then not a pure realization of an ONN, but rather a hybrid hardware. However, if the theoretical performance presented here is achievable in practice, then this is a technology very much worth pursuing, as there is a high demand for faster machine learning systems. Image classification is a good benchmark for assessing overall system potential.

The paper may impress a more accurate statement upon the reader by sooner making clear that the first set of results are not implemented in hardware, but rather simulated (if I understood correctly – no such clarification was made) and that performance drops notably upon actual hardware implementation (though instantiating such a system is still impressive!). There is also frequent language like “superiority” and “unprecedented,” without sufficient context. It would be helpful if, in one place, a concise characterization of energy and inference time is made as compared to the state-of-the-art in traditional hardware (besides calculations in supplementary material sections), as is

common practice in neuromorphic literature. There is table 1, for reference, but comparison is only made for VGG11/16 (not strongest networks) and the accuracy appears to be listed differently than what can be found online.

The paper could also benefit from increased clarity about the system itself. The actual mechanisms of learning could well be more elaborated. How exactly do the optical neurons share information with the artificial neurons? Some more background on ONNs and an explanation of terms like 'massive linear computations,' may increase readability.

Overall, the concept of "dual neurons" embodied in the DANTE system for training ONNs is a compelling case and merits further attention. This concept may extend beyond ONNs and into the broader context of difficult-to-train hardware. It is therefore a good thing that research of the kind presented here is being done.

**Response to referees**

Dear reviewers:

Thank you for your insightful comments on our manuscript titled “Training large-
scale optoelectronic neural networks with dual-neuron optical-artificial learning”. We
sincerely appreciate the reviewers’ positive feedback on the innovative aspects and their
valuable constructive comments, which have greatly contributed to enhancing the
manuscript’s quality. In the revised manuscript, we have made substantial revisions to
the content. Revised portion are marked in red in the manuscript and supplementary
information. The key modifications are as follows:

- 1) To increase the readability of our manuscript, we have rewritten our manuscript,
placing more emphasis on the physics, concepts, and interesting ideas. We have
revamped [**Fig. 1**] and rephrased the [**Results section Principle of dual-neuron**
**optical-artificial learning (DANTE)**]. We have provided background knowledge
on ANN and ONN to improve the readability, introduced the physical principles of
ONN concisely, discussed the challenges faced by existing approaches, and
explained why and how our proposed approach solve these challenges. (Reviewer
1, Reviewer 2, and Reviewer 3)
- 2) To highlight the advantages of DANTE, we have refined the results presented in
[**Fig. 2 and Fig. 3**], retaining only the essential findings that effectively showcase
our contributions, focusing more on explaining how and why our approach can
enable the training of large-scale ONNs which is previously impossible to train.
Details of the revisions are in [**Results section Improving ONN training using**
**DANTE**], [**Results section Large-scale ONNs enabled by DANTE**],
[**Supplementary Note S4 and Supplementary Table S1**]. (Reviewer 1 and
Reviewer 2)
- 3) To provide a comprehensive and clear explanation of how the ONN is trained using
DANTE, we have incorporated additional details on the training of the optical-
neuron layer using the impulse responses of the artificial-neuron layer. We have
introduced 3 subfigures within [**Fig. 2**] and refined the [**Results section**
**Improving ONN training using DANTE**], to illustrate the key training steps.
Additionally, we have included [**Supplementary Note S1 and Note S2**] and
[**Supplementary Fig. S1 and Fig. S2**], which present all the training details of
DANTE. (Reviewer 1, Reviewer 2, and Reviewer 3)

- 4) To accurately illustrate the purpose of the discussion on the number of operations
(OPs), we have refined the discussion regarding the comparison of parameters and
operations, placing greater emphasis on elucidating why our approach can achieve
approximately 100 times acceleration theoretically from the number of operations
(OPs). Details of the revisions are in [**Supplementary Note S6 and Table S2**].
(Reviewer 2)
- 5) We have concisely characterized the energy and inference time in [**Discussion**
**section**]. (Reviewer 3)
- 6) To make our manuscript more accessible and easier to understand, we have
provided background knowledge and clear explanations of the important terms.
Details of the revisions are in [**Fig. 1**] and the [**Results section Principle of dual-**
**neuron optical-artificial learning (DANTE)**]. (Reviewer 3)
- 7) We have made our code accessible on the Zenodo platform for reference
(<https://doi.org/10.5281/zenodo.8146362>). (Reviewer 2)
- We hope these modifications have significantly improved the quality and
readability of our manuscript, addressing the concerns raised by the reviewers.

**Reviewer #1:**

**R1-Q1.** I find the objective of the manuscript very interesting –if I have understand
correctly also the technique employed by the authors. I have reviewed the manuscript
at an earlier stage, and I must say that the authors have done some effort to improve the
readability. But it is still very cryptic. It is laden with a big amount of results, different
systems, numerous repetitions of the concept on a superficial level and its advantages.
I think I have grasped the method, but the authors still use convoluted concepts like
switches etc. All of this is done on 2 pages, constantly referencing why the here
presented approach is better, which is already stated in the introduction, the same
section and through the rest of the manuscript at various times. Why don't you just
explain concise what you do on half a page, then explain why it's better and than focus
on specific results. As it is I would suggest the authors rewrite the manuscript and focus
on the physics and interesting idea.

**A:** Thanks for the helpful comments. we have rewritten our manuscript, placing more
emphasis on the physics, concepts, and interesting ideas. We have revamped [Fig. 1]
and rephrased the corresponding paragraphs. We have provided background knowledge
on ANN and ONN, introduced the physical principles of ONN concisely, discussed the
challenges faced by existing approaches, and explained why and how our proposed
approach solve these challenges.

More precisely, the “switch” refers to a specific module within our dual-neuron
structure. This module is responsible for modifying the connectivity between the dual-
neuron optical-neuron layer and the artificial-neuron layer during the DANTE training
process. Through this network connection modification, we are able to optimize
parameters in a two-step manner. As demonstrated by Fig. 1d, in global artificial-
learning step, the dual-neuron switch connects all the artificial-neuron layers to form
an ANN with complex-valued linear computing layers. Global backpropagation is
utilized to end-to-end optimize them to fit the whole training dataset. In local optical-
learning step, the dual-neuron switch divides the network into independent layer pairs,
and connects the outputs of artificial- and optical- neuron layers to a loss function.
Within each layer pair, the trained artificial-neuron layer acts as a guide, and local
backpropagation is employed to optimize the parameters of the optical-neuron layer.
For more detailed information, please refer to [Supplementary Note S1-2 and Fig. S1-
2].

We also refined the results presented in [Fig. 2 and 3], retaining only the essential
findings that effectively highlight our advantages. Besides, we focused more on
explaining how and why our approach can enable the training of large-scale ONNs.

Main revision in the manuscript:

- 1) [Result section Principle of dual-neuron optical-artificial learning (DANTE)].
The entire section.
- 2) [Result section Improving ONN training using DANTE]. Figure 2abc
demonstrate the training process of the optical-neuron layer, ... please refer to
Supplementary Note 1 and Fig. S1.
- 3) [Result section Large-scale ONNs enabled by DANTE]. Figure 3e provides a
clear illustration of the spatial complexity ... please refer to Supplementary Note
S4 and Supplementary Table S1.

**R1-Q2.** Below please find some additional comments, but currently there are too many
things at odds with the manuscript for a detailed discussion of the text and results.
Physically relevant details are almost entirely left out or mentioned very late, and
instead the very most basic part of the principle idea is repeated again and again. There
are now 2 equations describing the concepts, but the conceptual part is still very weak
in its description.

**A:** Thank you for the feedback and suggestions. We have revamped [Fig. 1] and
rephrased the corresponding paragraphs. We have provided background knowledge on
ANN and ONN, introduced the physical principles of ONN concisely, discussed the
challenges faced by existing approaches, and explained why and how our proposed
approach solve these challenges. We also refined the results presented in [Fig. 2 and
Fig. 3], retaining only the essential findings that effectively highlight our advantages.

Main revision in the manuscript:

- 1) [Result section Principle of dual-neuron optical-artificial learning (DANTE)].
The entire section.
- 2) [Result section Improving ONN training using DANTE]. Figure 2abc
demonstrate the training process of the optical-neuron layer, ... please refer to
Supplementary Note 1 and Fig. S1.

**R1-Q3.** You: optimize a fully abstract and virtual ANN to perform a task in the first
stage, then you optimize an optical system to mirror that abstract ANN in a second stage,
and these stages are carried out for each learning epoch. Correct or not?

**A:** Thanks for the comments. In fact, it is not necessary to execute these two stages
during every learning epoch. [Supplementary Note S2 and Fig. S2] illustrates the
detailed steps of our dual-neuron optical-artificial learning approach:

Step 1: Initially, the switches connect all the artificial layers to generate a complexed-
valued ANN (denoted as ANN₀).

Step 2: Global artificial-learning is conducted to optimize the parameters of all the
connected artificial layers.

Step 3: Local optical-learning is carried out on the first un-learned optical-neuron layer,
closely matching it to its parallel artificial-neuron layer.

Step 4: If the matching error exceeds the threshold, the switches will connect the learned
optical-neuron layers and the artificial-neuron layers with un-learning optical-neuron
layers to generate a hybrid network. Global artificial-learning is then performed on the
connected artificial-neuron layers to finetune the network and compensate for the
introduced matching error. An exemplar hybrid network is illustrated in Supplementary
Fig. S1d, if the matching error of second optical modulation exceeds the threshold, the
first two optical-neuron layers and the remaining artificial-neuron layers are connected
to finetune the network.

Step 5: If the optical-neuron layer can match its parallel artificial-neuron layer well with
small enough error, we repeat step 3 on the next un-learned optical layer until all the
optical-neuron layers are optimized.

Step 6: Finetune the readout layer (if needed).

For simulation experiments based on the 4-f setup (Fig. 2 and Fig. 3), the artificial-
neuron layers can be carefully design so that they can be closely matched by the parallel
optical-neuron layers. As a result, both the global artificial-learning stage and local
optical-learning stage only need to be conducted once for all the layers. Generally
speaking, there are two cases that the ANN has to be finetuned (step 4).

1) Applying our trained optical modulation parameters to a physical system (Fig.
4). The physical assembly error, imperfect laser source, SLM modulation efficiency and
lenses introduce additional errors. Therefore, we have to finetune the ANN to
compensate for these errors. In the future, nanofabrication technology can be utilized
to solve the deployment error of ONNs.

2) The ONN system has to be updated for new data.

Supplementary Figure S2 | Step-by-step procedure of our dual-neuron optical-artificial learning (DANTE) approach. a, detailed steps of our dual-neuron optical-artificial learning (DANTE) approach. **b**, the dual-neuron network structure. Each layer consists of parallel an artificial-neuron layer (Art in figure) and an optical modulation layer (Opt in figure). The optical-neuron layer accurately models the optical diffraction based on Fourier optics. Meanwhile, the artificial-neuron layer aims to approximate the computationally intensive optical-neuron layer using easily-optimized complex-valued linear computing operations. **c**, the initial artificial neural network generated by connecting all the artificial-neuron layers, denoted as ANN₀. **d**, An exemplar hybrid artificial-optical neural network by connecting the first two optical-neuron layers and the remaining artificial-neuron layers. Backpropagation is conducted on the artificial-neuron layers to finetune the network.

**Reviewer #2:**

**R2-Q1.** In this manuscript, the authors described a training framework for optical
neural networks (ONNs). Instead of training the ONN using a digital model that
simulates the physical process of diffraction, they introduced an artificial neural
network (ANN) to model exactly the same forward computation to assist with training.
They first trained the ANN, since it involves fewer parameters, and then used the trained
ANN as the reference to training the physical model of ONN in a layer-by-layer fashion,
essentially by matching the impulse response of each layer of ONN and ANN.
Compared to direct training on the physical diffraction model, this method probably
improves training by reducing the computation of each layer to alleviate the vanishing
gradient problem, as the authors show in Fig. 2e and the associated discussion.

The manuscript tries to address an interesting and important problem, which is to find
a more efficient and reliable way to train the hardware of diffractive neural networks,
which at this stage still cannot be fully faithfully simulated on a computer to account
for intrinsic noise and error in an actual physical setup. Despite the proposed idea being
an interesting one, the work does not present enough evidence to support for the claim.

**A:** We thank the reviewer for the valuable feedback. Currently, the training of large-
scale wave-propagation-based ONNs encounters two primary challenges:

- 1) Fully faithfully simulating the wave-propagation process remains challenging due
to the intrinsic noise and error in an actual physical setup.
2) The substantial computational cost associated with the optical diffraction modeling.

Hence, in the revised manuscript, we provide the following extra evidences:

- 1) Theoretical analysis: We conducted a theoretical analysis of the training
acceleration accomplished by DANTE, from the aspect of the number of operations
(OPs) as detailed in [**Supplementary Note S6 and Table S2**]. The findings
demonstrate that the OPs associated with an optical-neuron layer exceed those of
an artificial-neuron layer by over 100 times. This observation provides a clear
explanation for the remarkable training acceleration.
- 2) Quantitative evaluation of computational cost: We measure the GPU memory cost
and time per epoch during the training of large-scale ONNs, as outlined in
[**Supplementary Note S4, Supplementary Table S1, and Figure 3e**]. Our results
indicated that utilizing existing direct optical diffraction modeling approaches, a 7-

layer ONN would demand over 200 GB of memory (batch size = 32) and several
195 weeks for completion. While with DANTE, the training could be completed within
196 two hours using only one single GPU.

3) Intrinsic noise and error: we have implemented an ONN fine-tuning method. This
approach ensured rapid adaptation of the trained ONN to the actual characteristics
of the physical system, as depicted in [Supplementary Note S1 and
Supplementary Fig. S1].

**R2-Q2.** It seems to me that the proposed training method would work, and it is an
improvement from the direct simulation of the physical process of diffraction in terms
of computation time and training effectiveness. However, most ONNs, including the 4f
convolution setup discussed extensively in this work (e.g., Fig. 1a), do not require
simulation of optical diffraction to start with, and this is only a specific technical
challenge with diffractive neural networks. The authors argued in a footnote that “the
frequential resolution of the FFT result may be different from the pixel pitch of the
physical devices”, and therefore the 4f setup cannot be efficiently modeled by FFT, but
has to be modeled by diffraction. I am not fully convinced by the statement, since
numerous prior works are training 4f convolutional neural networks without directly
simulating optical diffraction, for instance, Ref. 19, Ref. 18 (which is the same as Ref.
36 by mistake). To really make this point, the authors should compare their method to
other methods once used to train 4f systems. To be honest, in machine learning, there
is no absolute criteria on which training method is better or worse, and a lot of times, it
depends on the training hyperparameters and suitability to the application. Therefore, I
suggest authors focus on explaining how the training schemes solve a previously
impossible-to-solve training problem or how it might be helpful for other applications.

**A:** We sincerely appreciate the valuable comments provided by the reviewer. We
apologize for the erroneous conclusion drawn in the footnote. Both Ref 18 [1] and Ref
19 [2] also employ the 4-f setup in constructing their systems. In Ref 18 [1], amplitude
modulation is achieved using a digital micromirror device (DMD), whereas in Ref 19
[2], a phase mask is fabricated via photolithography for phase modulation. These
approaches employ different methods to address the size matching issue, rather than
relying on optical diffraction modeling alone. In Ref 18 [1], the authors resize and zero-
pad the input images from 28x28 (MNIST) or 32x32 (CIFAR) to 208x208 to properly
fit the DMD size (please refer to the supplement section 6 of [1]). In Ref 19 [2], the
authors fabricate a phase mask (known as a diffractive optical element (DOE)) through

multi-level photolithography, ensuring compatibility with the physical properties of the
DMD. Both methods demonstrate effective performance within their systems. In our
proposed approach DANTE, we model the optical diffraction step by step to solve the
physical size matching problem. Although this method requires more computations, it
offers the advantage of accommodating input images of varying sizes without resizing
and padding. This flexibility allows us to utilize off-the-shelf lenses with different focal
lengths and SLMs with varying pixel pitches.

As suggested, because the dual-neuron approach simultaneously alleviates the
burden of full physical model while retaining the modeling accuracy, our proposed
DANTE can solve a previously impossible-to-solve training problem and advance the
research in tackling large-scale practical problems. We have refined the results
presented in [Fig. 2 and Fig. 3], retained only the essential findings that effectively
highlight our advantages, focused more on how and why our approach can train large-
scale ONNs.

As presented in [Fig.3e, Supplementary Note S4, and Supplementary Table S1],
spatial and temporal complexity are two primary reason which makes large-scale ONNs
almost impossible-to-train by existing approaches. The single-neuron approach models
the entire network within the memory and optimizes all the phase masks simultaneously.
As depicted by the blue curve, the memory cost rises in proportion to the number of
trainable phase masks. As a result, training a 5-layer ONN with 12 phase masks at a
batch size of 8 would exceed the memory capacity (24 GB) of a single GPU. In contrast,
the artificial-neuron layer dramatically reduces spatial complexity, allowing the
network to be trained with little memory usage (< 6GB) even at a large batch size of
512. In the case of local optical learning, where only one phase mask needs to be trained
at a time, the memory cost keeps constant at approximately 2.5 GB. Furthermore, the
significant spatial complexity also implies a substantial temporal complexity. For
instance, on the CIFAR dataset, training a phase mask for one epoch takes
approximately 900 seconds for the single-neuron approach. Consequently, training the
7-layer ONN with 63 phase masks would require an average of 16 hours per epoch, and
completing the entire network training would require several weeks. However, with
DANTE, we can remarkably shorten the training process to a mere several hours. For
more details, please refer to Supplementary Note S4 and Supplementary Table S1.

Main revision in the manuscript:

1) [Result section Improving ONN training using DANTE]. Figure 2abc
demonstrate the training process of the optical-neuron layer, ... please refer to

Supplementary Note 1 and Fig. S1.

2) [Result section Large-scale ONNs enabled by DANTE]. Figure 3e provides a

clear illustration of the spatial complexity ... please refer to Supplementary Note

S4 and Supplementary Table S1.

3) **Supplementary Note S4 and Supplementary Table S1**

[1] Dalir, H. et al. Massively parallel amplitude-only Fourier neural network. *Optica*,

Vol. 7, Issue 12, pp. 1812-1819 7, 1812–1819 (2020).

[2] Chang, J., Sitzmann, V., Dun, X., Heidrich, W. & Wetzstein, G. Hybrid optical-

electronic convolutional neural networks with optimized diffractive optics for image

classification. *Scientific Reports* 2018 8:1 8, 1–10 (2018).

Figure 3e, GPU Memory cost for training these ONNs on CIFAR-10 dataset. bs, bias and scale. Mem., memory.

Supplementary Table S1. Memory cost and training time of large-scale ONNs on CIFAR-10 dataset. #, number; Mem., memory; bs, batch size.						
# of layers	3	4	5	6	7	10
# of phase masks	3	4	12	20	30	63
Existing approach (single-neuron)						
Mem. bs = 4 (GB)	2.89	3.16	13.36	20.82	27.42	61.05
Mem. bs = 32 (GB)	23.12	25.28	106.88	166.56	219.36	488.4
Training time / epoch (s)	2585.8	3480.2	10883.1 (~3 h)	18228.3 (~5 h)	28077.0 (~8 h)	58189.9 (~16 h)
DANTE (dual-neuron approach)						
Global artificial Mem. bs = 512	2.49	2.89	3.75	4.22	5.08	8.51

(GB)						
Global artificial Training time / epoch (s)	2.55	3.50	7.57	6.60	9.32	18.36
Local optical Mem. bs = 1 (GB)	2.36	2.36	2.36	2.36	2.36	2.36
Local optical Training time / phase mask (s)	218	218	218	218	218	218
Local optical All-layer training time (s)	654	872	2616	4360	6540	13734

**R2-Q3.** There is a lack of explanation on a very important step of the proposed training
 which is to compare the impulse response of an ANN layer to a delta function with that
 of an ONN layer. What training algorithm was used to optimize the parameters of the
 ONN layer? How many different delta functions are actually needed (despite the
 footnote saying “multiple”) Again, I strongly recommend the authors share their code
 online so that such details can be assessed from the reviewers’ point of view.

**A:** Thanks for the suggestion.

In the revised manuscript, we have incorporated additional details on the training
 process of the optical-neuron layer, specifically utilizing the impulse responses from
 the artificial-neuron layer. We introduced several subfigures within **[Fig. 2]** to illustrate
 the key training process. We also included **[Supplementary Note S1-2 and Fig. S1-2]**,
 which present comprehensive training details of DANTE.

In particular, Fig. 2abc demonstrate the training process of the optical-neuron layer,
 using a 4-f system as an example. Initially, the input is encoded into a coherent optical
 field, which propagates through two lenses and a trainable phase mask, producing
 outputs on the output plane. Based on Fourier optics, the 4-f based optical-neuron layer
 can be approximated as a complex-valued convolution operation with a large-size input
 plane and kernel. To further reduce the computational cost, it is decomposed into a
 multi-channel complex-valued convolutional operation with small-size inputs and
 kernels, serving as the forward function for the corresponding artificial-neuron layer.
 As the convolution operation is linear shift-invariant (LSI), the optical-neuron layer can
 be fully characterized by its 2D impulse response. The optimization process is
 demonstrated in Fig. 2c. A 2D Dirac delta function (impulse) is used as the input, and
 the resulting output is compared to the ground-truth label, which is obtained by

arranging the multi-channel complex-valued kernels into a tiled large-size single-
 channel complex-valued plane. ADAM-based backpropagation is employed to learn the
 optical modulation parameters. For more details on how the kernels and inputs are tiled,
 as well as the specifics of optimizing the optical modulation parameters, please refer to
 Supplementary Note 1 and Fig. S1.

Figure 2abc | Improving ONN training using DANTE. **a**, An optical-neuron layer, using a 4-f system as an example. **b**, The approximated artificial-neuron layer, featuring a multi-channel complex-valued convolutional operation. **c**, The training process of the optical-neuron layer.

For ONNs based on the 4-f system, we only need to fit one Dirac delta function,
 as the 4-f system is spatial-shift-invariant in the entire plane. For other types like
 diffractive deep neural networks (D²NN), it depends on the specific network settings.
 We have presented an example of extending DANTE to D²NN-like networks in
 [Supplementary Note S3 and Fig. S3]. We adopt Fourier Approximation to model the
 propagation between two surfaces into parallel small convolution operation. As the
 system is only partially spatial-shift-invariant, we optimize the phase mask to fit
 multiple impulses (3x3 here) with different spatial shifts, to enhance the spatial-shift-
 invariant property of the system.

Finally, as suggested, we have made our code accessible on the Zenodo platform
 for reference (<https://doi.org/10.5281/zenodo.8146362>).

Main revision in the manuscript:

1) [Result section Improving ONN training using DANTE]. Figure 2abc
 demonstrate the training process of the optical-neuron layer, ... please refer to

Supplementary Note 1 and Fig. S1.

2) **Supplementary Note S1 to S3, Supplementary Fig. S1 to S3**

**R2-Q4.** The manuscript claims “superior” performance on ImageNet and CIFAR-10
over digital models such as VGG. I support demonstrating ONNs on complex tasks
performed by state-of-the-art digital models. However, the claim is based solely on
simulation, and as far as I can see from the information provided, the simulation does
not consider noise or other intrinsic errors. The demonstration of an experimental setup
was only mentioned on line 271 and line 272. Most recent digital NN models can
achieve well above 90% accuracy on CIFAR-10 (>99% is not rare either), while the
study achieves 51% test accuracy. Therefore, I think it is unfair to claim ONNs achieve
better performance than digital NNs on hard tasks. For all similar claims in the paper, I
would like to see compelling evidence where an experimental demonstration achieves
higher test accuracy in CIFAR-10 or ImageNet compared to state-of-the-art digital
models, and some explanation on why before I can accept such “superiority” claims.

**A:** Thanks for the helpful comments. We acknowledge the gap that exists between
simulated ONNs and physically implemented ONNs. Compared to well-established
digital NN models, physical neural networks like ONNs do encounter the intrinsic error
and noise problem. On one hand, the gap may be shrunken with high-precision nano
fabrication, which is not the focus of this work. On the other hand, existing ONN could
already offer distinctive advantages, including high inference speed and exceptional
computational efficiency [1, 2]. However, it is worth noting that the considerable
computational cost associated with optical diffraction modeling has posed a hindrance
to the further scalability of optical neural networks, thus limiting their overall
performance potential. Hence, we proposed DANTE, aiming to reduce computational
costs, improve network convergence, and enable the successful training of large-scale
ONNs that are previously deemed impossible-to-train.

As you suggested, we have rewritten the manuscript to refine our claims and have
specifically revised the corresponding paragraphs of [Fig. 2 and Fig. 3]. We have also
included [Supplementary Note S6 and Table S2], providing a theoretical analysis of
the training acceleration accomplished by DANTE, and demonstrating that increasing
the scale of ONNs would lead to significant performance gain in simulation.

Main revision in the manuscript:

1) [Result section Improving ONN training using DANTE]. Figure 2abc

demonstrate the training process of the optical-neuron layer, ... please refer to
Supplementary Note 1 and Fig. S1.

2) **[Result section Large-scale ONNs enabled by DANTE]**. Figure 3e provides a
clear illustration of the spatial complexity ... please refer to Supplementary Note
S4 and Supplementary Table S1.

3) **Supplementary Note S6 and Table S2**

[1] Tiankuang Zhou, Xing Lin, Jiamin Wu, Yitong Chen, Hao Xie, Yipeng Li, Jingtao
Fan, Huaqiang Wu, Lu Fang, and Qionghai Dai. "Large-scale neuromorphic
optoelectronic computing with a reconfigurable diffractive processing unit." Nature
Photonics 15, no. 5 (2021): 367-373.

[2] Mario Miscuglio, Zibo Hu, Shurui Li, Jonathan K. George, Roberto Capanna,
Hamed Dalir, Philippe M. Bardet, Puneet Gupta, and Volker J. Sorger. "Massively
parallel amplitude-only Fourier neural network." Optica 7, no. 12 (2020): 1812-1819.

**R2-Q5.** The manuscript extensively discusses the number of parameters and the
number of operations of the models. However, it does not mention the equivalent bit
precision achieved by optical computing units. It is only fair to compare the number of
parameters and operations when the bit depth (or precision in the case of analog
computing) of the parameter and operation is known. Therefore, I do not find these
comparisons quite insightful or contributing to the study.

**A:** We express our gratitude to the reviewer for their valuable feedback. In our
experiments, we utilized 32-bit floating point numbers for training both DANTE and
the ANN (VGG and Wide Residual Network). Furthermore, in response to the
reviewer's comment, we have relocated this part of discussion to **Supplementary Note**
**S6 and Supplementary Table S2**, replaced FLOPs with the term "number of
operations" (OPs). Additionally, we have placed greater emphasis on elucidating why
our approach can achieve significant acceleration based on OPs and highlighted the
potential usefulness of this metric, rather than focusing on comparing ANN and ONN.
Specifically, the purpose of this discussion encompasses two main aspects:

1) Analyze the computing cost of training the artificial-neuron layer and optical-
neuron layer from theory. In DANTE, we approximate the optical-neuron layer into an
artificial-neuron layer, which consists of a complex-valued convolution operation.
Assume the convolution operation has an input size $N_i \times N_i$, input channel C_i , kernel
size N_k , and output channel C_o . Empirically, they should satisfy $(N_i + N_k) \times \lceil \sqrt{C_i} \rceil \times$

$\lceil \sqrt{C_o} \rceil \leq N_t/2$, where $N_t = 1200$ is the trainable phase mask size, and $(N_i + N_k) \times$
$\lceil \sqrt{C_i} \rceil \times \lceil \sqrt{C_o} \rceil$ approximately equals to the size of the tiled kernel. Exceeding this, the
local optical learning will introduce large errors, as the trainable phase mask size can
not cover enough frequencies on the Fourier plane. The number of MAD operations
(OPs) of this complex-valued convolution is $N_i^2 N_k^2 C_i C_o$. Because $(a + bj)(c +$
$dj) = (ac - bd) + j(ad + bc)$, a complex-number multiply operation needs 4 real-
number multiply operation and 2 real-number add operation (6 real-valued OPs). Thus,
a complex-valued multiply-add (MAD) operation requires 4 real-number multiply
operations and 4 real-number add operations, equals 8 real-valued OPs. The most
commonly used kernel size is $N_k = 3$, so the OPs of artificial-neuron layer is

$$409 \quad 8N_i^2 N_k^2 C_i C_o \approx N_k^2 (N_i \sqrt{C_i} \sqrt{C_o})^2 = N_k^2 \left(\frac{N_t}{2}\right)^2 = 25.92 \quad \text{MOPs} \quad (\text{real-valued}).$$

Conversely, in the optical-neuron layer, we model the optical diffraction step by step
for solving the size matching problem. The forward process needs 4 FFT, 4 inverse FFT
(iFFT), and 7 complex-valued element-wise multiplication. The OPs for FFT/iFFT are
around $5N^2 \log_2 N^2$ (real-valued), where N is the spatial size of simulation. Thus, the
OPs of an optical-neuron layer are $8 \times 5N^2 \log_2 N^2 + 6 \times 7N^2 = 3677.1$ MOPs
($N=2000$, real-valued), which is approximately $3677.1/25.9 \approx 140$ times of the
artificial-neuron layer. This explains why DANTE can achieve over 100 times
acceleration.

2) The artificial-neuron layer TOPs provides a more reliable metric of network
performance. As mentioned in the discussion section, different computing techniques
may have different definitions for counting the number of operations (OPs). Existing
ANNs usually use the FLOPs metric, which counts number of floating-point
calculations at given bit depth, such as FP32, FP16, etc. While current ONNs works
in analog domain, and the arithmetic types and sizes are usually limited. What is more,
wave-propagation-based ONNs such as diffractive neural networks are not designed to
complete traditional multiplication and addition operations. Miscuglio et al. only
compare the inference time between their proposed system and the GPU, without
discussing the OPs. Zhou et al. counts the number of optical connections between two
surfaces as the OPs, which models the optical diffraction as a fully-connected layer.
Therefore, directly comparing the OPs between ANN and ONN is unfair. Here, our
DANTE also gives another way to measure the performance of ONN. We have provided
the OPs and classification accuracies of our two 10-layer ONNs as well as the ANNs
used for comparison in **Supplementary Table S2**. It can be observed that the total OPs
of all the artificial-neuron layers serve as a reliable metric. When the OPs of the

artificial-neuron layers in an ONN are similar to the FLOPs of an ANN, their
 performances tend to be comparable as well. For example, 10-layer ONN vs. VGG11
 on the CIFAR-10 dataset, 10-layer ONN vs. VGG16 vs. WRN-1. While network with
 obviously higher OPs has significantly better performance, like WRN-2. WRN-k
 denotes the Wide Residual Networks, and k is the widening factor. For the VGG
 networks, since the input dimension of ImageNet-32 (32x32) is different from the
 original VGG input dimension (224x224), we reduce the size of the fully-connected
 layers and retrain the network. Hence the VGG FLOPs and performance in our table
 are different from that online.

Supplementary Table S2. the OPs and classification accuracies of our two 10-layer ONNs and the ANNs used for comparison.						
Dataset	CIFAR-10		ImageNet-32			
Network	10-layer ONN	VGG11	10-layer ONN	VGG16	WRN-1	WRN-2
Input dimension	28x28x3	28x28x3	32x32x3	32x32x3	32x32x3	32x32x3
FLOPs (ANN)	/	315.8 M	/	635.3 M	788.6 M	3.14 G
Real-valued OPs (Artificial-neuron layer)	441.0 M	/	808.7 M	/	/	/
# of phase masks	63	/	104	/	/	/
Real-valued OPs (Optical-neuron layer)	231.6 G	/	382.4 G	/	/	/
Accuracy (Top-1/top-5)	89.53%/	89.33%/	44.7%/	41.9%/	42.9%/	50.88%/
			69.0%	64.4%	67.5%	75.24%

Main revision in the manuscript:

1) **Supplementary Note S6 and Supplementary Table S2**

**R2-Q6.** Once more, I think the key message this manuscript tries to convey is a more
 robust and efficient training method for diffractive neural networks. I think the method
 proposed by the authors worked well in several aspects compared to the direct
 simulation of the physical process of diffraction; however, instead of focusing on
 explaining how this method might be useful for, the authors misplaced emphasis on
 several irrelevant claims that are not as solid if you think carefully about them.
 Nevertheless, this is still a good piece of work with some potential, and I would like to
 hear the response from the authors.

**A:** Thanks for the valuable comments. As you suggested, we have rewritten our

manuscript to refine these claims, and placed the emphasis on explaining how and why
our approach can enable the training of large-scale ONNs which is previously
impossible to train. Details of the revisions have been redlined in [**Results section**
*Improving ONN training using DANTE*], [**Results section Large-scale ONNs**
*enabled by DANTE*], [**Supplementary Note S4 and Supplementary Table S1**].

**Reviewer #3:**

**R3-Q1.** This paper presents a number of exciting results in a multi-faceted field using
an innovative training technique. Training optoelectronic neural networks (ONNs) is a
challenging task that demands careful solutions. While employing more conventional
artificial neural network (ANN) neurons in tandem with optical neurons is good idea,
it may be appropriate to more directly state that this is then not a pure realization of an
ONN, but rather a hybrid hardware. However, if the theoretical performance presented
here is achievable in practice, then this is a technology very much worth pursuing, as
there is a high demand for faster machine learning systems. Image classification is a
good benchmark for assessing overall system potential.

**A:** We appreciate the valuable feedback.

**R3-Q2.** The paper may impress a more accurate statement upon the reader by sooner
making clear that the first set of results are not implemented in hardware, but rather
simulated (if I understood correctly – no such clarification was made) and that
performance drops notably upon actual hardware implementation (though instantiating
such a system is still impressive!). There is also frequent language like “superiority”
and “unprecedented,” without sufficient context. It would be helpful if, in one place, a
concise characterization of energy and inference time is made as compared to the state-
of-the-art in traditional hardware (besides calculations in supplementary material
sections), as is common practice in neuromorphic literature. There is table 1, for
reference, but comparison is only made for VGG11/16 (not strongest networks) and the
accuracy appears to be listed differently than what can be found online.

**A:** We sincerely appreciate the reviewer's valuable comments.

We modified the corresponding paragraph of [**Fig. 2 and Fig. 3**], clearly
mentioned that the results are in simulation. We rewrote our manuscript to remove the

usage of languages without sufficient context. We provided a concise characterization
of energy and inference time in the **[Discussion section]** of the manuscript.

Regarding the performance drops observed in the actual hardware implementation,
we acknowledge the gap that exists between simulated ONNs and physically
implemented ONNs. On one hand, the gap may be shrunken with high-precision nano
fabrication, which is not the focus of this work. On the other hand, existing ONN could
already offer distinctive advantages, including high inference speed and exceptional
computational efficiency [1, 2]. However, it is worth noting that the considerable
computational cost associated with optical diffraction modeling has posed a hindrance
to the further scalability of optical neural networks, thus limiting their overall
performance potential. Hence, we proposed DANTE, aiming to reduce computational
costs, improve network convergence, and enable the successful training of large-scale
ONNs that are previously deemed impossible-to-train.

Regarding the issue raised in Table 1 of the original manuscript, we have
implemented the following changes. First, as recommended by R2, we have relocated
Table 1 to **[Supplementary Note S6]** and renumbered it as **[Supplementary Table S2]**.
Second, we have included two additional networks, namely WRN-1 and WRN-2, for
the purpose of comparison. Here, WRN-k refers to the Wide Residual Networks [3, 4],
where k represents the widening factor. Third, both our ONNs and the ANNs used for
comparison are evaluated using a downscaled version of the dataset known as
ImageNet-32 [4], which is constructed by resizing the ImageNet images to 32x32. To
fit the new input size, we modified the VGG networks. Hence, the accuracy appears to
be listed differently than what can be found online. The results presented in
Supplementary Table S2 demonstrates that the total OPs (operations) of all the artificial-
neuron layers serve as a reliable metric for accessing the network performance. When
the OPs of the artificial-neuron layers in an ONN closely align with the FLOPs
(floating-point operations) of an ANN, their performances are generally comparable.
Notably, networks with higher OPs have significantly better performance.

Main revision in the manuscript:

1) **[Discussion section]** Evaluating the computational performance, energy efficiency,
and inference time ... please refer to Supplementary Note S6.

2) **Supplementary Note S6 and Table S2**

[1] Tiankuang Zhou, Xing Lin, Jiamin Wu, Yitong Chen, Hao Xie, Yipeng Li, Jingtao
Fan, Huaqiang Wu, Lu Fang, and Qionghai Dai. "Large-scale neuromorphic

optoelectronic computing with a reconfigurable diffractive processing unit." *Nature*
*Photonics* 15, no. 5 (2021): 367-373.

[2] Mario Miscuglio, Zibo Hu, Shurui Li, Jonathan K. George, Roberto Capanna,
Hamed Dalir, Philippe M. Bardet, Puneet Gupta, and Volker J. Sorger. "Massively
parallel amplitude-only Fourier neural network." *Optica* 7, no. 12 (2020): 1812-1819.

[3] Zagoruyko, Sergey, and Nikos Komodakis. "Wide residual networks." *arXiv*
preprint arXiv:1605.07146 (2016).

[4] Patryk Chrabaszcz, Ilya Loshchilov, and Frank Hutter. "A downsampled variant of
imagenet as an alternative to the cifar datasets." *arXiv preprint arXiv:1707.08819*
(2017).

**R3-Q3.** The paper could also benefit from increased clarity about the system itself. The
actual mechanisms of learning could well be more elaborated. How exactly do the
optical neurons share information with the artificial neurons? Some more background
on ONNs and an explanation of terms like ‘massive linear computations,’ may increase
readability.

**A:** Thanks for the helpful suggestions. We have incorporated additional details on the
training process of the optical-neuron layer, specifically utilizing the impulse responses
from the artificial-neuron layer. We introduced several subfigures within **[Fig. 2]** to
illustrate the key training process. We also included **[Supplementary Note S1-2 and**
**Fig. S1-2]**, which present comprehensive details of DANTE.

In particular, Fig. 2abc demonstrate the training process of the optical-neuron layer,
using a 4-f system as an example. Initially, the input is encoded into a coherent optical
field, which propagates through two lenses and a trainable phase mask, producing
outputs on the output plane. Based on Fourier optics, the 4-f based optical-neuron layer
can be approximated as a complex-valued convolution operation with a large-size input
plane and kernel. To further reduce the computational cost, it is decomposed into a
multi-channel complex-valued convolutional operation with small-size inputs and
kernels, serving as the forward function for the corresponding artificial-neuron layer.
As the convolution operation is linear shift-invariant (LSI), the optical-neuron layer can
be fully characterized by its 2D impulse response. The optimization process is
demonstrated in Fig. 2c. A 2D Dirac delta function (impulse) is used as the input, and
the resulting output is compared to the ground-truth label, which is obtained by
arranging the multi-channel complex-valued kernels into a tiled large-size single-
channel complex-valued plane. ADAM-based backpropagation is employed to learn the
optical modulation parameters. For more details on how the kernels and inputs are tiled,

as well as the specifics of optimizing the optical modulation parameters, please refer to
 Supplementary Note 1 and Fig. S1.

To make our manuscript more accessible and easier to understand, we have
 provided background knowledge and clear explanations of the important terms. Details
 of the revisions are in [Fig. 1] and the [Results section Principle of dual-neuron
 optical-artificial learning (DANTE)].

Main revision in the manuscript:

1) [Result section Principle of dual-neuron optical-artificial learning (DANTE)].

The entire section.

2) [Result section Improving ONN training using DANTE]. Figure 2abc
 demonstrate the training process of the optical-neuron layer, ... please refer to
 Supplementary Note 1 and Fig. S1.

3) Supplementary Note S1 to S2, Supplementary Fig. S1 to S2

Figure 2abc | Improving ONN training using DANTE. **a**, An optical-neuron layer, using a 4-f system as an example. **b**, The approximated artificial-neuron layer, featuring a multi-channel complex-valued convolutional operation. **c**, The training process of the optical-neuron layer.

**Q4.** Overall, the concept of “dual neurons” embodied in the DANTE system for training
 ONNs is a compelling case and merits further attention. This concept may extend
 beyond ONNs and into the broader context of difficult-to-train hardware. It is therefore
 a good thing that research of the kind presented here is being done.

**A:** We thank the reviewer for the constructive feedbacks.

REVIEWER COMMENTS

Reviewer #1 (Remarks to the Author):

The manuscript reports on substantial improvements in (a) computing performance and (b) associated computational effort for the optimization process of optical neural networks.

The proposed approach is novel, the authors report substantial data on various benchmark tests and show that they can achieve an excellent degree of agreement between simulations and experiments. I find this an interesting and relevant contribution to the field and can therefore recommend publication.

I would like to thank the authors for the substantial effort of rewriting the explaining section. I would, however, suggest that they divide that section into several paragraphs. Currently it is 2 pages with I think only one or two paragraphs structuring the text - this makes it still hard to grasp the concept and to follow the thought process of the authors.

Reviewer #2 (Remarks to the Author):

I appreciate the authors for making detailed responses to my questions and concerns. The manuscript has been improved after revision regarding the clarity of technical details, such as what the training algorithm does. This new training algorithm contributes to the ONN field by adding a new training tool; therefore, I believe the work is worth publication. However, in this new version, there are still some claims made that are either ambiguous or inaccurate.

For example, in the abstract, the authors claim, " Unlike widely-used artificial neural networks (ANNs) targeting large-scale complex tasks, existing ONNs are still stuck in small network scales struggling with learning simple tasks since the complex mathematical modeling of the optical neurons is incompatible with the conventional 'end-to-end' optimization approaches inherited from ANNs." First, is the statement true by saying only simple tasks have been demonstrated on ONNs? Second, even if the statement is true, is it really because 'end-to-end' optimization cannot be implemented on large-scale ONNs? How confident are the authors in making a general statement like this on behalf of all different ONN architectures?

For another example, it says in the abstract, "Here, we innovate DANTE, a dual-neuron optical-artificial learning architecture, enabling the training of ONNs 20 times larger than existing models on the modern large-scale benchmark ImageNet dataset, which is an insurmountable task for existing single-neuron-based ONNs." It is not specified which models the authors compared to when claiming 20 times larger size. Did the authors really compare every single possible model published in the ONN field? I think the authors meant their previous work here. Also, Is the ONN presented in Fig. 4 really large enough to claim a complete triumph on training methods? The ONN might be wide so that it is relatively large, but is only 2-layer deep.

Another issue of this abstract is that there is no clear delineation on what conclusion is drawn from simulation and what from experiment. I strongly encourage the authors to revise it to make absolutely clear.

Reviewer #3 (Remarks to the Author):

The authors have done a nice job presenting results on the difficult task of training unconventional hardware by whatever means necessary. They have moreover improved the emphasis on when results are simulated, which is of particular importance. However, the abstract still blurs the line between simulation and reality, and also uses relative "improvement" results instead of absolute results, which may mislead the reader. This paper overall stands as a nice proof-of-concept that these two technologies may be beneficial to integrate in terms of information processing.

In general, these results apply to any work that may wish to combine digital machine learning with unconventional hardware. The dual-hardware approach might here be toted as a finding of its own.

The inclusion of more clear theory for training these networks has improved the readability of the work and bolstered understanding of the roll of traditional ANN methods have played in these results and that the ONN is just instantiating a mapping of ANN results to optical hardware.

I still find the methodology somewhat opaque, and organizationally hard to follow.

Because of the interdisciplinary nature of the work, (and because I am not an optics physicist), it is difficult to say if there is sufficient material here for the work to be reproduced.

Overall, visibility of this work appears to me as being of value to the AI-hardware community in general and I therefore expect it should be published.

Response to referees

Dear reviewers:

Thank you for your insightful comments on our manuscript titled “Training large-
scale optoelectronic neural networks with dual-neuron optical-artificial learning”. We
sincerely appreciate the reviewers’ positive feedback on the innovative aspects and their
valuable constructive comments, which have greatly contributed to enhancing the
manuscript’s quality. In the revised manuscript, we have made substantial revisions to
the content. Revised portion are marked in red in the manuscript and supplementary
information. The key modifications are as follows:

- 1) To increase the readability of our manuscript, we have rewritten the explaining
section [**Results section Principle of dual-neuron optical-artificial learning**
**(DANTE)**] and divided it into 5 paragraphs. (*Reviewer 1*)
- 2) In order to address the ambiguities and inaccuracies present in the [**Abstract**]
section, we have undertaken a thorough revision with the aim of presenting our
claims more clearly. We have clearly expressed the results and conclusion from
simulation and the physical experiment, respectively. Since the abstract section
should be unreferenced, when discussing the network size, we have removed the
relative claim “20 times larger” and used the absolute neuron number “150 million”
instead. In terms of the network prediction accuracy and network training time, we
have clearly stated the baseline methods. (*Reviewer 2 and Reviewer 3*)
- 3) To make the claims clear and accurate in the [**Introduction**] section, we have
revised the claims and placed them in the context of specific prior work. (*Reviewer*
*2*)
- 4) To ensure clarity and accuracy in physical experiment claims, we have revised our
manuscript. In the [**Abstract**] and [**Introduction**] sections, we have provided more
contents of the physical experiment and explicitly stated its objective (validating
physical feasibility). In the [**Result section DANTE on a physical ONN system**],
we have provided a comparison of training time between our system and the
previous study on the MNIST benchmark, demonstrating that DANTE can indeed
accelerate the physical ONN training. (*Reviewer 2*)
- 5) To enhance the reproducibility of our work, we have taken the initiative to expand
the content within the [**Section Supplementary Note S5**]. This extension provides
a more comprehensive guide, offering details into how to re-implement our

research in both simulation experiments and physical experiments. *(Reviewer 3)*
We hope these modifications have significantly improved the quality and
readability of our manuscript, addressing the concerns raised by the reviewers.

**Reviewer #1:**

**R1-Q1.** The manuscript reports on substantial improvements in (a) computing
performance and (b) associated computational effort for the optimization process of
optical neural networks. The proposed approach is novel, the authors report substantial
data on various benchmark tests and show that they can achieve an excellent degree of
agreement between simulations and experiments. I find this an interesting and relevant
contribution to the field and can therefore recommend publication.

**A:** We thank the reviewer for the positive feedbacks.

**R1-Q2.** I would like to thank the authors for the substantial effort of rewriting the
explaining section. I would, however, suggest that they divide that section into several
paragraphs. Currently it is 2 pages with I think only one or two paragraphs structuring
the text - this makes it still hard to grasp the concept and to follow the thought process
of the authors.

**A:** Thanks for the helpful comments. we have rewritten the explaining section and
divided it into 5 paragraphs.

1) In the 1st paragraph, we briefly review the artificial neural network (ANN) and
introduce the forward-pass mathematical model of the ANN layer.

2) The 2nd paragraph offers a concise overview of the diffraction-based ONN's
network structure, along with the forward-pass model of the ONN layer.

3) Moving to the 3rd paragraph, we provide the physical implementation details of the
ONN hardware system.

4) In the 4th paragraph, we review the network training and parameter optimization
methods employed by existing diffraction-based ONNs. We also address the challenges
posed by the high computational cost of optical-diffraction computations, which result
in difficulties in convergence and slow optimization speeds when training large-scale
ONNs.

5) Finally, in the 5th paragraph, we introduce our DuAl-Neuron opTical-artificial
lEarning (DANTE), including an explanation of the dual-neuron optical-artificial
network structure and the dual-neuron optical-artificial learning approach. We also
explain why DANTE has the potential to address the computing challenges and enable
the training of large-scale ONNs from a theoretical perspective.

Main revision in the manuscript:

1) **[Result section Principle of dual-neuron optical-artificial learning (DANTE)].**

The entire section.

**Reviewer #2:**

**R2-Q1.** I appreciate the authors for making detailed responses to my questions and
concerns. The manuscript has been improved after revision regarding the clarity of
technical details, such as what the training algorithm does. This new training algorithm
contributes to the ONN field by adding a new training tool; therefore, I believe the work
is worth publication.

**A:** We genuinely thank the reviewer for the positive comments.

**R2-Q2.** However, in this new version, there are still some claims made that are either
ambiguous or inaccurate. For example, in the abstract, the authors claim, “Unlike
widely-used artificial neural networks (ANNs) targeting large-scale complex tasks,
existing ONNs are still stuck in small network scales struggling with learning simple
tasks since the complex mathematical modeling of the optical neurons is incompatible
with the conventional ‘end-to-end’ optimization approaches inherited from ANNs.”
First, is the statement true by saying only simple tasks have been demonstrated on
ONNs? Second, even if the statement is true, is it really because ‘end-to-end’
optimization cannot be implemented on large-scale ONNs? How confident are the
authors in making a general statement like this on behalf of all different ONN
architectures?

**A:** We greatly appreciate your invaluable feedback. We apologize for the ambiguous
or inaccurate claims.

First, as far as our knowledge extends, there is no published ONN capable of
processing modern large-scale datasets like ImageNet, whether in simulations or
physical experiments. This is precisely why, in our previous abstract, we used the term
“simple tasks”. The underlying reason for this limitation lies in the restricted network
scale (both in width and depth) of existing ONNs. Second, for diffraction-based ONNs,
the high computational and memory cost associated with the optical diffraction
modeling pose significant challenges to the training of large-scale optical neural
networks. As presented in **[Supplementary Note S4 and Supplementary Table S1]**,
attempting to optimize all the optical neurons in a large-scale diffraction-based ONN
through end-to-end backpropagation demands extremely large GPU memory and
prolonged training times. Additionally, as demonstrated in Supplementary Fig. S7ab of

our prior research [1], increasing the network size (number of layers) may not always
yield performance improvements. This limitation arises due to the complexity of the
huge differentiable functions formed by connecting optical neurons, making them
challenging to optimize through end-to-end backpropagation. “End-to-end”
optimization of large-scale ONNs would become possible if these challenges can be
solved.

In the submitted manuscript, to ensure the clarity and accuracy of our claims, we
have made revisions to the abstract section: 1) We focused our scope to diffraction-
based ONNs. 2) We highlighted that the key challenge in training large-scale ONNs lies
in the computational and memory costs associated with optical diffraction modeling.
Specifically, we elucidate that existing diffraction-based ONNs attempt to directly
optimize all the computationally heavy optical neurons simultaneously using
backpropagation, resulting in slow training speed and convergence difficulty.

Supplementary Fig. S7. Ablation analysis of diffractive feedforward neural networks. The hyperparameters of the D²NN and D-NIN-1 models for the MNIST recognition, including the number of layers (a) and feature maps at each layer (b), were optimized during the in silico pre-training. The optimal number of layers was found to be 3 for both models, and the optimal number of feature maps at each layer was also 3 for the D-NIN-1 model. The recognition accuracy of the optimized D²NN and D-NIN-1 pre-trained models on the MNIST database was 97.6% and 98.8%, respectively. The recognition accuracies of D²NN on the MNIST database under different types of nonlinear activation functions are plotted in (c), where the incorporating of nonlinear functions electronically, i.e., ReLU, sigmoid, and binarization, in addition to the photoelectric nonlinearity don't or only slightly improve the accuracy.

Supplementary Fig. S7 in [1].

Main revision in the manuscript:

1) [Abstract] The entire section.

[1] Tiankuang Zhou, Xing Lin, Jiamin Wu, Yitong Chen, Hao Xie, Yipeng Li, Jingtao
Fan, Huaqiang Wu, Lu Fang, Qionghai Dai. “Large-scale neuromorphic
optoelectronic computing with a reconfigurable diffractive processing unit.”
Nature Photonics, vol. 15, no. 5, 2021.

We revised the abstract to: “... *However, training large-scale diffraction-based ONNs*
*face challenges due to the high computational and memory costs of optical diffraction*
*modeling, resulting in slow training speed and convergence difficulty. Here, we*
*innovate DANTE, a dual-neuron optical-artificial learning architecture. Optical*
*neurons model the optical diffraction, while artificial neurons approximate the*
*intensive optical-diffraction computations with lightweight functions. Unlike existing*
*single-neuron learning method that directly connects and optimizes all the*
*computationally heavy optical neurons simultaneously through backpropagation,*
*DANTE decouples the optical neurons by employing iterative global artificial-learning*
*steps and local optical-learning steps, leading to better and faster convergence. ...”.*

**R2-Q3.** For another example, it says in the abstract, “Here, we innovate DANTE, a
dual-neuron optical-artificial learning architecture, enabling the training of ONNs 20
150 times larger than existing models on the modern large-scale benchmark ImageNet
dataset, which is an insurmountable task for existing single-neuron-based ONNs.” It is
not specified which models the authors compared to when claiming 20 times larger size.
Did the authors really compare every single possible model published in the ONN field?
I think the authors meant their previous work here.

**A:** We sincerely appreciate your valuable feedback. We apologize for the ambiguous or
inaccurate claims. We present the parameters of representative large-scale diffraction-
based ONN models in the following table, encompassing ONNs both in simulation and
in physical experiments:

Model	# of layers	# of masks	# of neurons per mask	# of neurons
D-NIN-1 [1]	3	7	700×700	3.6 M
Residual-D ² NN [2]	30	30	200×200	1.2 M
Ensemble of D ² NN N = 30 [3]	5	5×30	200×200	0.2 M × 30 = 6 M
Ensemble of D ² NN N = 77 [3]	5	5×77	200×200	0.2 M × 77 = 15.4 M

5-layer D ² NN [4]	5	5	300×300	0.45 M
10-layer D ² NN [4]	10	10	300×300	0.9 M
Fourier-space D ² NN (cell) [5]	5	5	800×800	3.2 M
Fourier-space D ² NN (CIFAR-10) [5]	10	10	160×160	0.26 M
Amplitude-only Fourier neural network [6]	1	16	832×832	11.08 M
D ² NN at Visible Wavelengths [7]	5	5	1000×1000	5.0 M
Programmable D ² NN [8]	5	5	8x8	320
Multiscale diffractive U-Net [9]	11	11	128×128 (max) layer different	0.12 M
Ours (10-layer ImageNet)	10	104	1200x1200	149.8 M

To the best of our knowledge, our diffraction-based ONN model stands out as the
largest in terms of the number of neurons, boasting approximately about 150-M neurons.
In contrast, others models typically range from 0.1-M to 20-M neurons. The ensemble
of N=77 D²NN model features 15.4-M neurons [2], but the model is constructed
through an ensemble learning approach involving 1252 pre-trained 5-layer ONNs. Each
small 5-layer ONN is independently trained with 0.2-M neurons. The amplitude-only
Fourier neural network comprises 11 M neurons [6], but its depth is shallow with just
one optical-computing layer. Our ONN model is both wide (neurons per layer) and deep,
crucial for enhancing network learning ability[9, 10]. Compared to our previous study
[1], the ONN trained in this manuscript is 20 times larger.

Delving further into the discussion, training a large-scale ONN faces a twofold
challenge (in simulation): 1) In the case of diffraction-based ONNs, the computational
and memory demands associated with optical diffraction modeling are substantial
hurdles to effective network training. As presented in **[Supplementary Note S4 and**
**Supplementary Table S1]**, attempting to optimize all the optical neurons in a large-
scale diffraction-based ONN through end-to-end backpropagation demands extremely
large GPU memory and long training times. 2) The optimization of huge differentiable
functions formed by cascading optical neurons is difficult. Direct end-to-end learning
via backpropagation often results in poor convergence. This issue is exemplified in
Supplementary Fig. S7ab of our prior research [1], increasing the network size (number
of layers) may not always yield performance improvements. Hence, training approach
is also crucial for successfully training large-scale ONNs.

In order to rectify the ambiguities in our claims, we have made revisions to both
the abstract and introduction sections of the manuscript. In the introduction section, we
placed the claim of network size comparison in the context of specific prior works.
Since the abstract section should be unreferenced, we removed the “20 times larger”
network size claim and used the absolute neuron number instead.

Main revision in the manuscript:

- 1) **[Abstract]** The entire section.
- 2) **[Introduction]** The 4th paragraph.
- 3) **[Supplementary Table S3 | Network scale of existing large-scale ONNs our**
**DANTE]** new table.

- [1] Tiankuang Zhou, Xing Lin, Jiamin Wu, Yitong Chen, Hao Xie, Yipeng Li, Jingtao
Fan, Huaqiang Wu, Lu Fang, Qionghai Dai. “Large-scale neuromorphic
optoelectronic computing with a reconfigurable diffractive processing unit.”
Nature Photonics, vol. 15, no. 5, 2021.
- [2] Hongkun Dou, Yue Deng, Tao Yan, Huaqiang Wu, Xing Lin, Qionghai Dai.
“Residual D2NN: training diffractive deep neural networks via learnable light
shortcuts.” Optics Letters, vol. 45, no. 10, 2020.
- [3] Md Sadman Sakib Rahman, Jingxi Li, Deniz Mengü, Yair Rivenson, Aydogan
Ozcan. “Ensemble learning of diffractive optical networks.” Light: Science &
Applications, vol. 10, no. 1, 2021.
- [4] Tao Yan, Jiamin Wu, Tiankuang Zhou, Hao Xie, Feng Xu, Jingtao Fan, Lu Fang,
Xing Lin, Qionghai Dai. “Fourier-space diffractive deep neural network.” Physical
review letters, vol. 123, no. 2, 2019.
- [5] Mario Miscuglio, Zibo Hu, Shurui Li, Jonathan K. George, Roberto Capanna,
Hamed Dalir, Philippe M. Bardet, Puneet Gupta, Volker J. Sorger. “Massively
parallel amplitude-only Fourier neural network.” Optica, vol. 7, no. 12, 2020.
- [6] Hang Chen, Jianan Feng, Minwei Jiang, Yiqun Wang, Jie Lin, Jiubin Tan, Peng Jin.
“Diffractive deep neural networks at visible wavelengths.” Engineering, vol. 7, no.
10, 2021.
- [7] Che Liu, Qian Ma, Zhang Jie Luo, Qiao Ru Hong, Qiang Xiao, Hao Chi Zhang,
Long Miao et al. “A programmable diffractive deep neural network based on a
digital-coding metasurface array.” Nature Electronics, vol. 5, no. 2, 2022.
- [8] Yiming Li, Zexi Zheng, Ran Li, Quan Chen, Haitao Luan, Hui Yang, Qiming
Zhang, Min Gu. “Multiscale diffractive U-Net: a robust all-optical deep learning
framework modeled with sampling and skip connections.” Optics Express, vol. 30,
no. 20, 2022.
- [9] Zhou Lu, Hongming Pu, Feicheng Wang, Zhiqiang Hu, Liwei Wang. “The
expressive power of neural networks: A view from the width.” Advances in neural
information processing systems, 2017.
- [10] Thao Nguyen, Maithra Raghu, Simon Kornblith. “Do wide and deep networks
learn the same things? uncovering how neural network representations vary with
width and depth.” arXiv preprint, 2020.

We revised the abstract to: “... *In simulation experiments, DANTE successfully*
*trains large-scale ONNs with 150 million neurons on the modern ImageNet benchmark,*
*achieving performance on par with the representative VGG network, which is*
*previously unattainable. It also accelerates training speeds by two orders of magnitude*
*on the CIFAR-10 benchmark compared to the single-neuron learning approach. ...”.*

**R2-Q4.** Also, Is the ONN presented in Fig. 4 really large enough to claim a complete
triumph on training methods? The ONN might be wide so that it is relatively large, but
is only 2-layer deep.

**A:** We deeply appreciate the invaluable feedback provided. Although the full simulation
results are not presented in experiments, the proof-of-concept physical experimental
results do validate the core of DANTE’s feasibility. We achieved a 96% accuracy on
the MNIST dataset, comparable to our previous study [1]. But the training time was
significantly reduced from more than 5 hours to about 445 seconds, indicating that
DANTE can indeed accelerate the physical ONN training. On the other hand, we
acknowledge that there is still a gap between the simulated ONNs and the physically
implemented ONNs due to the intrinsic error and noise in physical system. But the
optical results still significantly outperform the baseline method, which proves that our
physical ONNs can effectively extract features from the input images. The performance
gap may be further narrowed by integrating high-precision nanomanufacturing, which
is not the focus of this work.

To ensure clarity and accuracy in our claims, we have made revisions in the
submitted manuscript. In the **[Abstract]** and **[Introduction]** sections, we clearly
express the results and conclusion from the simulation and the physical experiment,
respectively. We also provided more contents of the physical experiment and explicitly
state its objective (validating physical feasibility). In the **[Result section DANTE on a**
**physical ONN system]**, we provided a comparison of training time between our system
and the previous study [1] on the MNIST benchmark.

Main revision in the manuscript:

1) **[Abstract]** The entire section.

2) **[Introduction]** The 4th paragraph.

3) **[Result section DANTE on a physical ONN system]** The 3rd paragraph.

[1] Tiankuang Zhou, Xing Lin, Jiamin Wu, Yitong Chen, Hao Xie, Yipeng Li, Jingtao
Fan, Huaqiang Wu, Lu Fang, Qionghai Dai. “Large-scale neuromorphic
optoelectronic computing with a reconfigurable diffractive processing unit.”
Nature Photonics, vol. 15, no. 5, 2021.

We revised the abstract to: “... *In simulation experiments, DANTE successfully*
*trains large-scale ONNs with 150 million neurons on the modern ImageNet benchmark,*
*achieving performance on par with the representative VGG network, which is*
*previously unattainable. It also accelerates training speeds by two orders of magnitude*
*on the CIFAR-10 benchmark compared to the single-neuron learning approach. In*
*physical experiments, we develop a two-layer ONN system capable of effectively*
*extracting features to enhance the classification of natural images, serving as a*
*validation of DANTE’s physical feasibility. ...”.*

**R2-Q5.** Another issue of this abstract is that there is no clear delineation on what
conclusion is drawn from simulation and what from experiment. I strongly encourage
the authors to revise it to make absolutely clear.

**A:** Thank you for the valuable feedback. We have rewritten the [**Abstract**] and
[**Introduction**] section to clearly express the results and conclusion from simulation
and the physical experiment, respectively.

Main revision in the manuscript:

1) [**Abstract**] The entire section.

2) [**Introduction**] The 4th paragraph.

We revised the abstract to: “... *In simulation experiments, DANTE successfully*
*trains large-scale ONNs with 150 million neurons on the modern ImageNet benchmark,*
*achieving performance on par with the representative VGG network, which is*
*previously unattainable. It also accelerates training speeds by two orders of magnitude*
*on the CIFAR-10 benchmark compared to the single-neuron learning approach. In*
*physical experiments, we develop a two-layer ONN system capable of effectively*
*extracting features to enhance the classification of natural images, serving as a*
*validation of DANTE’s physical feasibility. ...”.*

**Reviewer #3:**

**R3-Q1.** The authors have done a nice job presenting results on the difficult task of
training unconventional hardware by whatever means necessary. They have moreover
improved the emphasis on when results are simulated, which is of particular importance.
However, the abstract still blurs the line between simulation and reality, and also uses
relative "improvement" results instead of absolute results, which may mislead the
reader.

**A:** We sincerely value the invaluable feedback you've provided. We have rewritten the
abstract to clearly express the results and conclusion from simulation and the physical
experiment, respectively. More specifically, in terms of network size, we have replaced
the previous relative descriptor, "20 times larger", with the absolute size "150 million
neurons". Also, we selected the representative VGG network as the baseline for
benchmarking the ImageNet classification performance. In terms of the network
training time, we have explicitly stated that the training acceleration achieved by
DANTE is measured against the existing single-neuron learning method. The single-
neuron learning method directly connects and optimizes all the computationally heavy
optical neurons simultaneously through backpropagation.

Main revision in the manuscript:

- 1) **[Abstract]** The entire section.
2) **[Introduction]** The 4th paragraph.

We revised the abstract to: "... *Unlike existing single-neuron learning method that*
*directly connects and optimizes all the computationally heavy optical neurons*
*simultaneously through backpropagation, DANTE decouples the optical neurons by*
*employing iterative global artificial-learning steps and local optical-learning steps,*
*leading to better and faster convergence. In simulation experiments, DANTE*
*successfully trains large-scale ONNs with 150 million neurons on the modern ImageNet*
*benchmark, achieving performance on par with the representative VGG network, which*
*is previously unattainable. It also accelerates training speeds by two orders of*
*magnitude on the CIFAR-10 benchmark compared to the single-neuron learning*
*approach. In physical experiments, we develop a two-layer ONN system capable of*
*effectively extracting features to enhance the classification of natural images, serving*

*as a validation of DANTE's physical feasibility. ...".*

**R3-Q2.** This paper overall stands as a nice proof-of-concept that these two technologies
may be beneficial to integrate in terms of information processing. In general, these
results apply to any work that may wish to combine digital machine learning with
unconventional hardware. The dual-hardware approach might here be toted as a finding
of its own. The inclusion of more clear theory for training these networks has improved
the readability of the work and bolstered understanding of the roll of traditional ANN
methods have played in these results and that the ONN is just instantiating a mapping
of ANN results to optical hardware. I still find the methodology somewhat opaque, and
organizationally hard to follow. Because of the interdisciplinary nature of the work,
(and because I am not an optics physicist), it is difficult to say if there is sufficient
material here for the work to be reproduced.

**A:** We appreciate the valuable feedback.

We have made our code for simulation experiments available, which is
implemented using PyTorch. All the necessary information can be found in the source
code, covering aspects such as optical diffraction modeling, optical modulation
elements, artificial neuron approximation, network structure definition, and the
optimization modules for DANTE.

For the simulation experiment, we have open-sourced DANTE code. The code is
implemented it using PyTorch with GPU support. All the necessary information can be
found in the source code, covering aspects such as optical diffraction modeling, optical
modulation elements modeling, artificial neuron definition, optical neuron definition,
network structure definition, and the two-step learning algorithm. These classes,
optical-neuron layer and artificial-neuron layer, have been created as extensions of the
torch.nn.Module class. Their usage closely mirrors that of PyTorch ANN layers like
torch.nn.Linear and torch.nn.Conv2d. Researchers and engineers familiar with PyTorch
programming can readily employ these functions and classes to either re-implement our
research or construct their own networks.

For physical experiments, the optical devices employed are detailed in the Method
section. The optical diagram and real system image are provided in Fig. 4. The SLM
calibration curve is shown in Supplementary Fig. S9a, and the network structures are
demonstrated in Supplementary Fig. S10. The step-by-step procedure of physical
experiments is outlined in Supplementary Note S5. The optical components and
elements utilized in our system, including the laser, square aperture, lens, polarizer,

beam splitter, and the spatial light modulator (SLM), are all widely recognized and
commonly used in optical experimental setups. An experienced optics engineer can
reimplement can calibrate our system within just a few days.

Finally, to enhance the reproducibility of our work, we have taken the initiative to
expand the content within the [**Section Supplementary Note S5**]. This extension
provides a more comprehensive guide, offering details into how to re-implement our
research in both simulation experiments and physical experiments.

Main revision in the manuscript:

1) [**Section Supplementary Note S5**] we expanded this section.

**R3-Q3.** Overall, visibility of this work appears to me as being of value to the AI-
hardware community in general and I therefore expect it should be published.

**A:** We truly appreciate the reviewer for the positive remarks.

REVIEWERS' COMMENTS

Reviewer #1 (Remarks to the Author):

I thank the authors for their hard work and recommend publication.

Reviewer #2 (Remarks to the Author):

I appreciate the authors for responding to my concerns and making corresponding changes. The manuscript has been much improved to better convey the key messages of the manuscript, and overall I would support its publication.

In the revised abstract, the authors used diffraction-based ONNs to refer to diffractive neural networks. I think it would be more precise to just use diffractive neural networks, since it refers to a specific architecture, while most optical neural networks, including some on-chip ones are based on diffraction, which have not been studied in this work. This is just a subtle semantic difference. Toward the end in discussion, the authors may discuss how their methods can be extended to other optical-neural-network architecture.

Reviewer #3 (Remarks to the Author):

The authors have met all requested criteria to merit publication. Their diligence is appreciated, and their future work is happily anticipated.

**Response to referees**

Dear reviewers:

Thank you for your insightful comments on our manuscript titled “Training large-
scale optoelectronic neural networks with dual-neuron optical-artificial learning”. We
sincerely appreciate the reviewers’ positive feedback on the innovative aspects and their
valuable constructive comments, which have greatly contributed to enhancing the
manuscript’s quality. In the revised manuscript, we have made substantial revisions to
the content. Revised portion are marked in red in the manuscript and supplementary
information. The key modifications are as follows:

1) To make our claims more precise, we changed “diffraction-based ONNs” to
“diffractive neural networks” in our revised manuscript. Also, we have added a
paragraph to the discussion section, discussing the extension of our methods to
other optical neural network architectures. (*Reviewer 2*)

We hope these modifications have significantly improved the quality and
readability of our manuscript, addressing the concerns raised by the reviewers.

**Reviewer #1:**

**R1-Q1.** I thank the authors for their hard work and recommend publication.

**A:** We thank the reviewer for the positive feedbacks.

**Reviewer #2:**

**R2-Q1.** appreciate the authors for responding to my concerns and making
corresponding changes. The manuscript has been much improved to better convey the
key messages of the manuscript, and overall I would support its publication.

**A:** A: We sincerely appreciate the reviewer for their favorable remarks.

**R2-Q2.** In the revised abstract, the authors used diffraction-based ONNs to refer to
diffractive neural networks. I think it would be more precise to just use diffractive
neural networks, since it refers to a specific architecture, while most optical neural
networks, including some on-chip ones are based on diffraction, which have not been
studied in this work. This is just a subtle semantic difference. Toward the end in
discussion, the authors may discuss how their methods can be extended to other optical-
neural-network architecture.

**A:** Thank you for your valuable feedback. We have made revisions to our manuscript,
changing “diffraction-based ONNs” to “diffractive neural networks” in our revised
manuscript. Additionally, we have added an extra paragraph to the discussion section,
discussing the extension of our methods to other optical neural network architectures.

*We have added the following paragraph in the **Discussion** section: “While we*
*currently focus on demonstrating the performance of DANTE on 4-f system-based*
*diffractive neural networks, the underlying concept can be extended to other types of*
*diffractive neural networks as well. The implementation details and results regarding*
*this extension are presented in Supplementary Note S3 and Fig. S3. In the future, we*
*can further broaden the applicability of our method to encompass other types of ONN*
*architectures, such as on-chip diffraction-based ONNs and integrated chip diffractive*
*neural network. Through the utilization of optical-artificial dual neurons in modeling*

*ONN chips, we have the potential to expedite network training and increase the size of*
*trainable networks.”*

**Reviewer #3:**

**R3-Q1.** The authors have met all requested criteria to merit publication. Their diligence
is appreciated, and their future work is happily anticipated.

**A:** We genuinely thank the reviewer for the positive comments.